

# Assessing the Added Value of the Intermediate Complexity Atmospheric Research Model (ICAR) for Precipitation in Complex Topography

Johannes Horak[1], Marlis Hofer[1], Fabien Maussion[1], Ethan Gutmann[2], Alexander Gohm[1], and Mathias W. Rotach[1]

[1]Universität Innsbruck, Department of Atmospheric and Cryospheric Sciences, Innsbruck, Austria
[2]Research Applications Laboratory, National Center for Atmospheric Research, Boulder, Colorado, USA

**Correspondence:** Johannes Horak (johannes.horak@uibk.ac.at)

**Abstract.** The coarse grid spacing of global circulation models necessitates the application of climate downscaling to investigate the local impact of a changing global climate. Difficulties arise for data sparse regions in complex topography which are computationally demanding for dynamic downscaling and often not suitable for statistical downscaling due to the lack of high quality observational data. The Intermediate Complexity Atmospheric Research Model (ICAR) is a physics-based model that can be applied without relying on measurements for training and is computationally more efficient than dynamic downscaling models. This study presents the first in-depth evaluation of multi-year precipitation time series generated with ICAR on a $4 \times 4 \, \mathrm{km}^2$ grid for the South Island of New Zealand for the eleven-year period from 2007 to 2017. It focuses on complex topography and evaluates ICAR at 16 weather stations, eleven of which are situated in the Southern Alps between $700 \, \mathrm{m} \, \mathrm{MSL}$ and $2150 \, \mathrm{m} \, \mathrm{MSL}$. ICAR is diagnosed with standard skill scores and the effect of model top elevation, topography, season, atmospheric background state and synoptic weather patterns on these scores are investigated. The results show a strong dependence of ICAR skill on the choice of the model top elevation, with the highest scores obtained for $4 \, \mathrm{km}$ above topography. Furthermore, ICAR is found to provide added value over its ERA-Interim reanalysis forcing data set for alpine weather stations, improving mean squared errors (MSE) up to $53\,\%$. It performs similarly during all seasons with an MSE minimum during winter, while flow of higher linearity and atmospheric stability were found to increase skill scores. ICAR scores are highest during weather patterns associated with flow perpendicular to the Southern Alps and lowest for flow parallel to the alpine range. While measured precipitation is underestimated by ICAR, these results show the skill of ICAR in a real-world application, and may be improved upon by further observational tuning or bias correction techniques. Based on these findings ICAR shows the potential to generate downscaled fields for long term impact studies in data sparse regions with complex topography.

*Copyright statement.* TEXT



## 1   Introduction

Global circulation models (GCM) generate atmospheric datasets on spatiotemporal grids that, especially in complex topography, are too coarse to investigate the local impact of a changing global climate. To bridge the gap between local and GCM scales, a variety of downscaling methods and techniques exist (Christensen et al., 2007), roughly characterizable as dynamic downscaling (e.g. Hill, 1968; Rasmussen et al., 2014), statistical downscaling (e.g. Klein et al., 1959; Benestad et al., 2008) or as intermediate complexity downscaling (e.g. Sarker, 1966; Smith and Barstad, 2004; Gutmann et al., 2016).

While dynamic downscaling results in a self-consistent set of atmospheric fields, the computational cost required for the fine spatial and temporal grid spacing is high, especially for long-term simulations or sensitivity studies. The drawback of statistical downscaling is the associated requirement of high quality measurements for model training, rendering it less applicable to data sparse regions. Even more problematic, as soon as observation-based training or tuning is applied, the assumption of stationarity is introduced for statistical downscaling and, to a lesser extent, to dynamic downscaling as well, which may not hold under a changing climate (Maraun, 2013; Gutmann et al., 2012). Both classes are, overall, therefore not ideally suited for the long-term study of the regional effects of a changing global climate. These problems are particularly amplified in glacierized areas, which are often located in hard-to-access, remote regions and complex topography. For such regions weather station deployment and maintenance is often impractical or too expensive, resulting in a scarcity of continuous measurements and inapplicability of statistical downscaling approaches. In case of dynamic downscaling the correct representation of the influence of complex topography on local weather and climate leads to a high computational cost. This cost is further increased by the long response times of glaciers to climatic changes, which are on the order of several decades (Raper and Braithwaite, 2009). Process-based glacier models therefore require long-term information about the state of the atmosphere above the glacier to investigate the impact of a changing global climate.

The Intermediate Complexity Atmospheric Research model (ICAR; Gutmann et al., 2016) offers a computationally frugal and physics-based alternative that does not rely on measurements with the linear theory of orographic precipitation as its theoretical foundation. In comparison to other downscaling approaches of intermediate complexity (e.g. Sarker, 1966; Rhea, 1977; Smith and Barstad, 2004; Georgakakos et al., 2005), ICAR is a more general atmospheric model that requires fewer simplifying assumptions about the state of the atmosphere, such as spatial and temporal homogeneity of the background flow.

At the time of writing, ICAR has been evaluated in an idealized hill experiment, as well as by comparing monthly precipitation fields generated by ICAR for Colorado, USA, with WRF output and an observation-based gridded dataset (Gutmann et al., 2016). Furthermore, ICAR was employed to generate downscaled atmospheric fields as input for a glacier mass balance model to simulate meltwater runoff in the western Himalayas (Engelhardt et al., 2017). Recently Bernhardt et al. (2018) applied ICAR to investigate differences in precipitation patterns and amounts for a domain in the European Alps due to the choice of the microphysics scheme and associated parameters. However, Gutmann et al. (2016) evaluated ICAR for season totals and based on one year of precipitation data, while Bernhardt et al. (2018) only investigated a 7 month period.

This study conducts the first multi-year evaluation of ICAR, and compares ICAR precipitation fields to data from individual weather stations in different terrains. As a starting point for investigating the added value of ICAR, New Zealand is chosen. Here





the precipitation regime is strongly orographically influenced by the Southern Alps (Sturman and Wanner, 2001). The island is isolated from major land masses and moist air from the surrounding ocean is advected toward the orographic ridge of the Southern Alps at a predominantly right angle. Measurements from 16 weather stations within the study domain, eleven of which

are alpine stations located in complex topography, are used to quantify added value with regards to ERA-Interim interpolated to station location. Furthermore the model performance is diagnosed with respect to season, background atmospheric state and synoptic weather patterns. Average and seasonal precipitation patterns are compared to an operational gridded rainfall data set. Additionally, the influence of the choice of the model top height onto the downscaled results is discussed.

## 2 ICAR - description and setup

### 2.1 Overview

ICAR (Gutmann et al., 2016) is a three-dimensional atmospheric model based on linear mountain wave theory, which predicts the wind field based on the topography and the background state of the atmosphere (Sawyer, 1962; Smith, 1979). Within this transient wind field, ICAR numerically advects atmospheric quantities, such as heat and moisture as supplied by a forcing dataset at the model boundaries. In its standard setup ICAR employs the Thompson microphysical scheme (Thompson et al.,

2008), a double moment scheme in cloud ice and rain, to compute the mixing ratios of water vapor, cloud water, rain, cloud ice, graupel and snow.

The classic approach of linear theory assumes a steady and horizontally uniform stable background state with first order perturbations caused by, for instance, the topography. It does not take into account interactions among waves or waves and turbulence, nor transient phenomena such as time-varying wave amplitudes, gravity wave breaking or low level flow splitting.

A basic discussion of the limitations implicit to these assumptions can be found in Nappo (2012). Additionally, especially for large domains, a uniform background state may not be an acceptable approximation.

ICAR overcomes the latter drawback by calculating the perturbations for a predefined combination of atmospheric background states for the entire domain and storing the results in a lookup table. Then, for each volume element in the domain, the perturbation corresponding to its background state, which is given by the forcing data set, is selected from this table by

interpolating between the closest matches in the lookup table. This assembly routine is carried out for every forcing timestep, yielding a time sequence of steady state wind fields between which ICAR interpolates linearly. A detailed description of this process is given in Gutmann et al. (2016). To avoid unstable atmospheric condition present in the forcing dataset or caused by the microphysics, ICAR enforces stability by ignoring imaginary values of the Brunt-Väisälä frequency and substituting them with a minimum positive value of $10^{-7}\mathrm{s}^{-1}$. In the version of ICAR employed in this study, the reflection of mountain waves at the interface of atmospheric layers is neglected.



## 2.2 Model setup

ICAR can be run without relying on measurements for observation-based tuning. Therefore, it is of particular interest for data-absent, mountainous or glacierized regions (e.g. Pepin et al., 2015). This study aims at quantifying a baseline performance of

ICAR with default settings as it would be applied for a region where no data are available. For individual sites, improvement is then possible by using data based tuning, as routinely performed in regional climate model based downscaling. However, the model top of ICAR could not be adopted from default settings (Horak et al., 2018), see Sect. 2.3. The ICAR configuration used in this study (configuration file available as download, see Horak et al., 2018) employs the wind field computation process as described in Sect. 2.1 and by Gutmann et al. (2016), an upwind advection scheme to transport quantities within the wind

field and the Thompson microphysics scheme. Coupling between the surface and the atmosphere is neglected, i.e., no turbulent surface fluxes of heat, moisture and momentum are considered. Atmospheric fields were downscaled to a $4 \times 4\,\mathrm{km}^2$ horizontal grid and to an hourly time step.

## 2.3 Model top

For dynamic downscaling models the position of the model top is a critical parameter. A higher model top implies, in principal, a more faithful representation of atmospheric processes and physics that in turn leads to an increased computational cost while a lower setting has the opposite effect. In light of these requirements the ICAR default setting of $5.7\,\mathrm{km}$ above topography as used in Gutmann et al. (2016) is comparatively low. Preliminary studies indicated that for a model top at $5.7\,\mathrm{km}$ only a small added value can be obtained for the South Island of New Zealand. Additionally, these preliminary studies showed that different

choices for the model top elevation influenced the precipitation patterns and amounts throughout the study domain, leading to significant changes in model skill. With the sensitivity study described in Sect. 4.3 the optimal elevation of the model top for this study was determined as $4\,\mathrm{km}$ above topography.

## 2.4 Digital elevation model

The model domain in this study, as depicted in Fig. 1, encompasses the entire South Island of New Zealand and a small section

of the North Island. The digital elevation model (DEM) employed was upscaled from the $1' \times 1'$ ETOPO1 Ice (Amante and Eakins, 2009) DEM to $4 \times 4\,\mathrm{km}^2$, corresponding to $205 \times 225$ gridpoints. Since peaks represented by only one grid point increase the wave energy in the high frequency part of the spectrum, leading to unphysical atmospheric perturbations, the topography was smoothed by a $3 \times 3$ moving window algorithm (Guo and Chen, 1994, p.34). A similar type of smoothing, which is common when using the weather research and forecasting pre-processing system, was performed in previous studies employing ICAR (Gutmann et al., 2016; Engelhardt et al., 2017).





## 2.5 Forcing data and reference

In this study, ERA-Interim reanalysis data (ERAI; Dee et al., 2011) are used as the forcing data set for ICAR. Reanalysis data are obtained from computationally expensive state-of-the-art general circulation model re-forecasts constrained by quality-

controlled observations with a variational data assimilation procedure. Therefore, reanalysis data are of a particular relevance for data-scarce high mountain regions around the globe, where the application of solely interpolation-based gridded historical data sets is problematic. ERAI have a horizontal grid spacing of $0.75°\times0.75°$ (globally), corresponding to approximately $81\times110\,\mathrm{km}^2$ within the study domain (defined in Sect. 3), and extends to the $0.1\,\mathrm{hPa}$ pressure level in the vertical. The assembled ICAR forcing file contains ERAI zonal and meridional winds $U$ and $V$, potential temperature $\Theta$, pressure $p$, specific

humidity $q_v$, and surface pressure $p_0$ at each $6\,\mathrm{h}$ h forcing time step and every grid point within the domain.

ERA interim reanalysis data were also used as ICAR forcing in the study of Bernhardt et al. (2018). Bernhardt et al. (2018), however, evaluated only the precipitation sum over a seven month period. They emphasize the importance of mountain weather station networks to allow for a more detailed evaluation of ICAR. Gutmann et al. (2016) used the North American Regional Reanalysis (NARR), which has a $32\,\mathrm{km}$ grid spacing (Mesinger et al., 2006). Engelhardt et al. (2017) use output from the

Norwegian Earth System Model (NorESM), downscaled to a grid spacing of $25\,\mathrm{km}$ by the regional climate model REMO, as ICAR input for a simulation period from 2006 to 2099. In this study, ERAI are preferred over regional reanalysis data sets because of their global availability and thus more widespread applicability.

## 2.6 Convective precipitation

The ICAR configuration for this study, as described in Sect. 2.2, is able to model orographic precipitation and, at least in part,

precipitation driven by the synoptic scale. To account for convective precipitation, convective precipitation from the ERAI reanalysis, $P_{\mathrm{CP}}$, is resampled to the ICAR timestep and bilinearly interpolated in space to the sites of interest and then added to the ICAR precipitation time series $P_{\mathrm{I}}$:

$$P(t) = P_{\mathrm{I}}(t) + P_{\mathrm{CP}}(t), \tag{1}$$

where in the following the $P(t)$ time series is referred to as ICAR$_{\mathrm{CP}}$ and $P_{\mathrm{I}}(t)$ as ICAR.



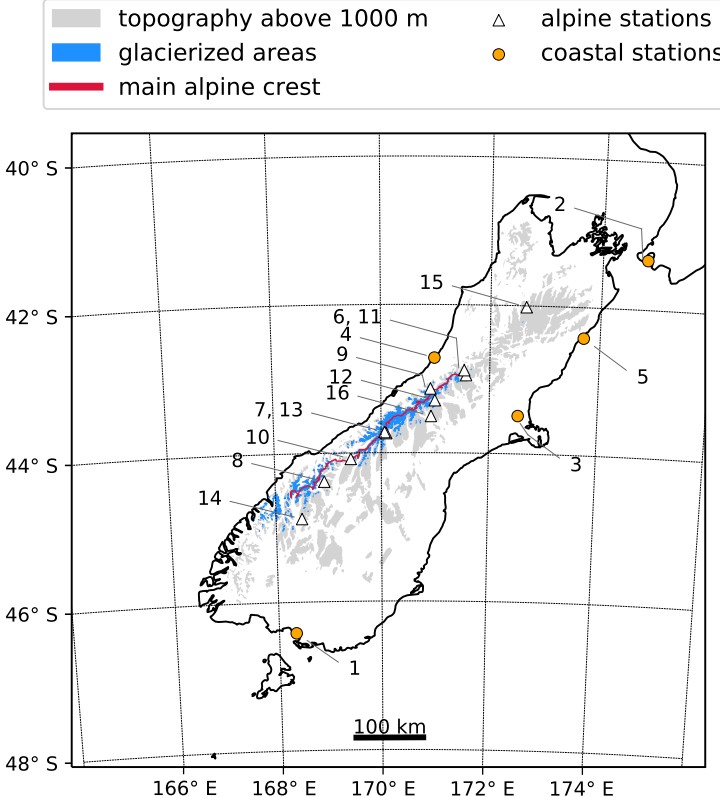

**Figure 1.** The South Island of New Zealand. Shown are the coast (black line), the topography above an elevation of $1000\,\mathrm{mMSL}$ (gray shading), glacierized areas (blue shading) and the approximate location of the main alpine crest (red line). The alpine weather stations considered in the evaluation of this study are indicated by white triangles, while coastal weather stations are represented by orange disks. The numbers next to the markers are ordered from lowest to highest weather station elevation and may be used to look up additional information for each station in Table 1.

## 3   Study Domain and Observational Data

### 3.1   Overview

5   This study focuses on the Southern Alps on the South Island of New Zealand located in the southwestern Pacific Ocean. The mountain range is oriented southwest-northeast and runs almost parallel to the western coast of the island. It is approximately $800\,\mathrm{km}$ long and $60\,\mathrm{km}$ wide, extends across a latitude range from $41\,^\circ\mathrm{S}$ to $46\,^\circ\mathrm{S}$ and consists of a series of ranges and basins (Barrell et al., 2011). Of New Zealand's $3144$ glaciers with a surface area larger than $10^{-2}\,\mathrm{km}^2$, all except for $18$ lie within the Southern Alps (Chinn, 2001). The domain and glacierized areas are depicted in Fig. 1.



New Zealand's climate is characterized as humid and maritime with prevailing westerly winds. The average precipitation patterns are influenced by the Southern Alps, which act as a topographic barrier for these moist winds (Chinn, 2001). The resulting orographically influenced precipitation regime is characterized by a precipitation maximum of up to $14\,\mathrm{m\,yr^{-1}}$ on the

western flanks close to the Main Divide of the Southern Alps. Along the west coast on average $5\,\mathrm{m\,yr^{-1}}$ of precipitation are observed while the plains east of the Main Divide receive less than $1\,\mathrm{m\,yr^{-1}}$ (Griffiths and McSaveney, 1983; Henderson and Thompson, 1999). Additionally, the strong westerly winds in the Southern Alps may lead to significant spillover, distributing precipitation to leeward slopes (Chater and Sturman, 1998)

## 3.2 Obervational Data

Precipitation time series from the weather stations in complex topography were supplied by the National Institute of Water and Atmospheric Research of New Zealand (NIWA) and the University of Otago, New Zealand (Cullen and Conway, 2015). In case of the coastal weather stations, records from the New Zealand National Climate Database (NCD) were employed. The individual time series extend over an eleven year period, with the shortest time series covering $0.8$ years and the longest $11$ years. Details concerning the weather stations, accumulated annual precipitation and time series length are listed in Table 1.

Furthermore, Table 1 includes an average downwind distance $\Delta$ from the main alpine crest of the Southern Alps. It is calculated with regards to westerly and northwesterly flow, the wind directions associated with the largest mean precipitation, see Sections 4.7 and 4.8.

Different instruments were employed to measure rainfall at the weather stations in the study region. At Christchurch, Invercargill and Kaikoura precipitation measurements were carried out with a tipping-bucket rain gauge, while different gauges were

employed at the remaining coastal stations: A standard rain gauge at Hokitika and a drop gauge at Wellington. Precipitation at Mount Brewster was measured with a tipping-bucket rain gauge and data post-processing is described in detail by Cullen and Conway (2015). Cullen and Conway (2015) identified the period for reliable precipitation data at the site as extending approximately from the end of December until the end of April, during which it was adjusted for gauge undercatch. Outside of this period, Cullen and Conway applied a scaling function to extrapolate from rain gauge data at a site 30km southwest

of Brewster Glacier at $320\,\mathrm{m}$ MSL. Precipitation at the alpine NIWA stations was measured with tipping-bucket rain gauges. Heating systems were not installed, however, a wind shield was in place at Mueller Hut. The raw data available from the NCD is provided by the Meteorological Service of New Zealand, NIWA and, in three cases, unidentified observing authorities. For this study, all NIWA and NCD input data were subject to basic plausibility checks. They identified and flagged data points exceeding 20 standard deviations from the mean, with negative values, or with excessive temporal persistence. Marked entries

were then manually reviewed and removed from the dataset if physically unreasonable values were found. The thus quality controlled data were then used for further processing and resampled to daily accumulated precipitation $P_{24h}$. Days that had gaps in their original time series were not considered for further analysis. The number of missing days is documented in Table 1.

To compare simulated precipitation patterns across the South Island of New Zealand to an observational dataset, the NIWA virtual climate station gridded daily rainfall product (VCSR; Tait and Turner, 2005) is employed. The VCSR is an observation based data set interpolated to a horizontal grid spacing of $3'$ or approximately $5\,\mathrm{km}$. It scales rainfall at high elevations and



**Table 1.** List of weather stations used in this study sorted by their elevation. The table lists station number, elevation $z$, latitude (lat), longitude (lon), name, average distance downwind of the main crest of the Southern Alps ($\Delta$) based on westerly and northwesterly flow, mean annual precipitation $\bar{P}$, fraction of convective precipitation in ERAI annual sum $f_{cp}$, length of the time series ($l$) and number of days removed due to missing entries or failed quality checks ($d_m$). The superscript following the station name indicates the data provider: NCD (1), NIWA (2) and University of Otago (3). Precipitation data for Larkins and Potts were lineary extrapolated to a full year. $\Delta$ was not considered for coastal weathers stations and no values were assigned for Mahanga and Larkins since they lie north respectively south of the main alpine crest.

| No. | $z$ (m MSL) | lat (°) | lon (°) | Name | $\Delta$ (km) | $\bar{P}$ (m yr$^{-1}$) | | | $f_{cp}$ (1) | $l$ (yr) | $d_m$ (d) |
|---|---|---|---|---|---|---|---|---|---|---|---|
| | | | | | | measured | ICAR$_{\text{CP}}$ | ERAI | | | |
| Coastal Stations | | | | | | | | | | | |
| 1 | 0 | -46.42 | 168.33 | Invercargill[1] | | $1.0 \pm 0.1$ | $1.7 \pm 0.1$ | $1.1 \pm 0.1$ | 0.47 | 11.0 | 1 |
| 2 | 4 | -41.33 | 174.80 | Wellington[1] | | $0.8 \pm 0.1$ | $1.1 \pm 0.1$ | $0.8 \pm 0.1$ | 0.67 | 11.0 | 4 |
| 3 | 37 | -43.49 | 172.53 | Christchurch[1] | | $0.5 \pm 0.1$ | $0.8 \pm 0.1$ | $0.7 \pm 0.1$ | 0.78 | 11.0 | 1 |
| 4 | 39 | -42.72 | 170.98 | Hokitika[1] | | $2.8 \pm 0.2$ | $3.2 \pm 0.3$ | $1.6 \pm 0.2$ | 0.29 | 4.7 | 3 |
| 5 | 105 | -42.42 | 173.70 | Kaikoura[1] | | $0.6 \pm 0.1$ | $1.1 \pm 0.1$ | $0.7 \pm 0.1$ | 0.80 | 5.8 | 68 |
| Alpine Stations | | | | | | | | | | | |
| 6 | 738 | -42.95 | 171.57 | Arthur Pass[2] | 7 | $4.4 \pm 0.5$ | $2.3 \pm 0.2$ | $1.3 \pm 0.1$ | 0.27 | 6.5 | 9 |
| 7 | 765 | -43.74 | 170.10 | Aoraki/Cook[2] | 7 | $4.1 \pm 0.6$ | $2.8 \pm 0.3$ | $1.5 \pm 0.1$ | 0.17 | 10.5 | 2 |
| 8 | 1280 | -44.38 | 168.93 | Albert Burn[2] | 11 | $2.9 \pm 0.2$ | $1.6 \pm 0.2$ | $2.0 \pm 0.2$ | 0.28 | 3.2 | 2 |
| 9 | 1390 | -43.13 | 170.91 | Ivory[2] | -2 | $7.3 \pm 0.5$ | $5.7 \pm 0.8$ | $1.6 \pm 0.1$ | 0.41 | 6.4 | 22 |
| 10 | 1650 | -44.08 | 169.43 | Brewster[3] | 0 | $6.0 \pm 0.4$ | $2.4 \pm 0.2$ | $1.7 \pm 0.1$ | 0.15 | 5.3 | 10 |
| 11 | 1655 | -42.88 | 171.53 | Philistine[2] | 0 | $4.8 \pm 0.6$ | $4.1 \pm 0.4$ | $1.4 \pm 0.1$ | 0.42 | 5.4 | 6 |
| 12 | 1752 | -43.29 | 171.00 | Raikai[2] | 12 | $2.1 \pm 0.2$ | $2.3 \pm 0.2$ | $1.5 \pm 0.1$ | 0.32 | 6.7 | 10 |
| 13 | 1818 | -43.72 | 170.06 | Mueller Hut[2] | 3 | $5.1 \pm 1.2$ | $3.4 \pm 0.4$ | $1.6 \pm 0.1$ | 0.34 | 3.2 | 10 |
| 14 | 1925 | -44.88 | 168.49 | Larkins[2] | - | 1.1 | 1.0 | 1.9 | 0.30 | 0.8 | 2 |
| 15 | 1955 | -42.02 | 172.65 | Mahanga[2] | - | $2.2 \pm 0.1$ | $1.8 \pm 0.1$ | $1.2 \pm 0.1$ | 0.40 | 5.3 | 14 |
| 16 | 2128 | -43.50 | 170.93 | Potts[2] | 35 | 2.0 | 0.9 | 1.6 | 0.41 | 0.9 | 2 |





remote locations using data from mesoscale model simulations. While the VCSR does not necessarily represent the actual distribution of precipitation (Tait et al., 2012), and may miss precipitation events (Tait and Turner, 2005), it serves as an approximation to an observational gridded dataset and is based on observations and physics-based regional climate modeling.

## 4 Methods and Results

### 4.1 Evaluation Strategy

In this study, $\text{ICAR}_{\text{CP}}$ time series (see Sect. 2.6) are evaluated in terms of the added value over total precipitation from the ERAI reanalysis. Added value in this context is used as in the investigation of regional climate model based downscaling, where it is defined as the comparative performance of the regional climate model output to the global driving data (e.g. Di Luca et al., 2015). The aim is not a downscaling method intercomparison (e.g. ICAR versus WRF; Gutmann et al., 2016). Similar studies with a focus on quantifying the added value over the driving input have been performed for full dynamic downscaling (for a review see Torma et al., 2015). This way, our study serves as guidance whether at all, and, if so, under which conditions ICAR can add value over ERAI with a particular focus on complex terrain.

The available data are grouped by selected criteria that are expected to affect the added value, in particular the topographic complexity, seasons, flow linearity and the synoptic situation. Flow linearity is characterized by the inverse non-dimensional mountain height, in the following referred to as Froude number, calculated for test volumes upstream of the weather stations. The synoptic situation is determined by weather patterns as employed in an operational weather pattern classification scheme.

### 4.2 Skill Scores and Significance Test

Mainly two scores are employed to quantify the added value of $\text{ICAR}_{\text{CP}}$ over ERAI: the mean squared error (MSE) based skill score $\text{SS}_{\text{MSE}}$ and the Heidke skill score HSS. The MSE based skill score (Wilks, 2011b, Chapter 8) is given by

$$\text{SS}_{\text{MSE}} = 1 - \frac{\text{MSE}}{\text{MSE}_r}, \tag{2}$$

where MSE is the MSE of $\text{ICAR}_{\text{CP}}$ $P_{24\text{h}}$ and $\text{MSE}_r$ is the MSE of $P_{24\text{h}}$ of the reference model (here, ERAI). This way, $\text{SS}_{\text{MSE}}$ can be interpreted as percentage improvement (reduction of error) by $\text{ICAR}_{\text{CP}}$ relative to ERAI.

The contingency table based Heidke Skill score (HSS; Wilks, 2011b, Chapter 8) is used to analyze events that are characterized by either their occurrence or absence, such as, for instance, $P_{24\text{h}}$ exceeding a given threshold, and whether the tested model is able to correctly diagnose the occurrences in comparison to a reference model. Thresholds investigated in this study are $1\,\text{mm}$, $25\,\text{mm}$ and $50\,\text{mm}$ for $24\,\text{h}$ accumulated precipitation. HSS is defined as

$$\text{HSS}(r) = \frac{r - r_r}{1.0 - r_r} \tag{3}$$

where $r$ is the proportion correct of $\text{ICAR}_{\text{CP}}$ and $r_r$ that of the ERAI reference model. The proportion correct is given by $r = (a + d)/n$ with $n = a + b + c + d$. In this context $a$ is the amount of times the event was forecast and observed to occur





(hits), $b$ the number of events that were forecast but not observed (false hits), $c$ the number of events that were not forecast but observed (false alarm or missed event), $d$ the amount of times an event was neither forecast nor observed (correct misses) and the total number of cases $n$.

The scores defined by equations 2 and 3 both yield values in the interval $(-\infty, 1]$ and condense the information on whether the tested model performs better with respect to a skill measure than a reference model into one number. A model exactly reproducing the measurements corresponds to a score of $1$, a score of $0$ is achieved if the model performs equally well as the reference model, and lower scores are found if the model is outperformed by the reference model.

Moving block bootstrap (MBB) is employed to determine the significance of the skill scores (Wilks, 2011a, Chapter 5). The

procedure is similar to ordinary bootstrapping with the distinction that, instead of $n$ individual observations, blocks of length $L$ are resampled. For the time series considered in this study values of $L$ range between 1 and 9, with the autocorrelation structure of the time series preserved within each block, and different blocks independent of each other. Each skill score is recalculated for $10\,000$ MBBs of the original data, yielding a sampling distribution of the respective score. If the fifth percentile of this distribution is positive, the score obtained from the original time series is considered significant.

## 4.3    Model top sensitivity study

The results of a sensitivity study used to determine the optimal position of the model top are summarized in Fig. 2. Simulations for six different model top elevations were run for a two year reference period (2014-2015) and the MSE was calculated for the ICAR and ICAR$_{CP}$ time series at all alpine weather stations. The reference period was chosen as the time slice when a maximum of observational data was available, with measured time series for nine out of eleven alpine weather stations (except

for Potts and Larkins) being available during this period. The model top setting yielding the lowest average MSE for the alpine stations was considered optimal.

The lowest average MSE for ICAR was found for a model top elevation of $2.5\,\mathrm{km}$ above topography, while for ICAR$_{CP}$ the minimum is at $4.0\,\mathrm{km}$, see Fig. 2. Setting the model top higher or lower quickly deteriorates model performance for ICAR and ICAR$_{CP}$ alike. Potential reasons for the observed behavior are discussed in Sect. 5. Furthermore, the sensitivity analysis

indicates that the majority of skill is already present in the ICAR time series. Nonetheless, the inclusion of ERAI convective precipitation, as described in Sect 2.6, results in an additional reduction in MSE for the ICAR$_{CP}$ time series at all simulated model top settings. The results are similar when, instead of the mean MSE, the mean $\mathrm{SS_{MSE}}$ is investigated (not shown). The mean skill maxima for ICAR and ICAR$_{CP}$ are again found at $2.0\,\mathrm{km}$ and $4.5\,\mathrm{km}$ respectively, with ICAR$_{CP}$ showing the highest mean skill of $0.24$. All following analyses, unless stated otherwise, therefore focus on the ICAR$_{CP}$ time series obtained with a model top set to $4.0\,\mathrm{km}$ above topography.




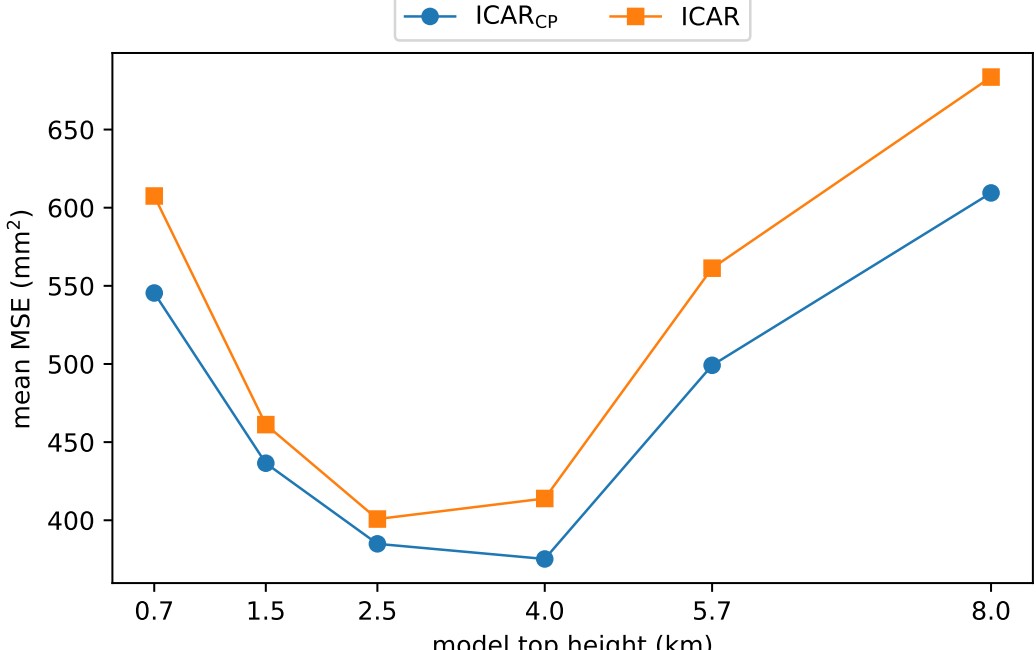

**Figure 2.** The average MSE of ICAR and ICAR$_{CP}$ time series from simulations for the reference period 2014-2015 at alpine weather stations as a function of the chosen model top (in km above topography). Connecting lines serve as guides to the eye.

## 4.4 Overall Performance of ICAR for Alpine and Coastal Weather Stations

The performance of ICAR$_{CP}$ at individual stations is presented in Table 2 and summarized in Fig. 3. For the alpine weather stations, values of SS$_{MSE}$ calculated across the entire period when data is available (see Table 1 for details) indicate a median SS$_{MSE}$ of 0.3, equivalent to a 30% reduction of error on median relative to ERAI for locations in complex alpine topography. Six out of eleven alpine stations have significant scores above zero, three are negative. Regarding the topographic situation (see Fig. 1), six alpine weather stations are downwind of the main alpine ridge, with respect to the predominant wind directions. The results indicate a negative correlation between SS$_{MSE}$ and the average distance downwind to the main alpine crest ($\Delta$), with the weather stations farthest leeward (Albert Burn, Raikai and Potts) exhibiting, apart from Mahanga, the lowest scores observed. No $\Delta$ value was assigned to Mahanga since it is located to the north of the alpine crest and situated approximately $80\,\mathrm{km}$ downwind from the coast. The topography to its west and northwest up until the coast is constituted by scattered mountain ranges with elevations between $1000\mathrm{m}$ and $1800\mathrm{m}$.

In terms of HSS at alpine stations, median scores above $0.14$ are found for the $P_{24h}$ thresholds $25\mathrm{mm}$ and $50\mathrm{mm}$ respectively, see Fig. 3b. The only weather stations with comparatively large negative scores are Mahanga and Raikai, the former of which is located downstream of mountainous terrain and the latter the second farthest downwind of the main alpine crest. For days





with $P_{24h}$ exceeding 1mm significant added value of ICAR$_{CP}$ over ERAI is only found at two out of eleven locations. Since only small negative scores are found and the median score is $0.01$ for all alpine stations, this indicates, that at this threshold ICAR$_{CP}$ performs very similar to ERAI, and that ICAR$_{CP}$ does not improve on modeling the occurrence of precipitaton.

Table 2 contains additional information about the relative abundance of threshold exceedances at each weather station. The performance of individual stations is discussed separately in Sect. 5.

A direct comparison of measured and simulated $P_{24h}$ time series at the alpine stations Albert Burn and Ivory is shown in Fig. 4. These two sites were selected since among alpine stations for the entire period SS$_{MSE}$ is lowest at Albert Burn and highest at Ivory. During the year 2015 (second half shown) the skill difference is largest, with SS$_{MSE}$ yielding $-0.39$ and $0.58$

respectively. The two weather stations are separated by a distance of about $210\,km$ and at almost the same elevation, with Albert Burn at $1280\,m\,MSL$ and Ivory at $1390\,m\,MSL$. However, Albert Burn is located $11\,km$ downstream of the main alpine ridge, while Ivory lies approximately 2km upstream of it according to the definition in Sect. 3.2. At both sites ICAR reproduces the features of the measured precipitation time series, but in the case of Albert Burn it underestimates measured precipitation amounts on average by almost $50\%$ and, even at Ivory where ICAR performs best in terms of SS$_{MSE}$, precipitation is still

underestimated by approximately $22\%$. The potential factors contributing to the observed underestimations are discussed in Sect. 5.

Figure 3b shows that for the coastal weather stations, no added value could be found as quantified by SS$_{MSE}$ and HSS for the thresholds $P_{24h} > 25\,mm$ and $P_{24h} > 50\,mm$. Slightly positive values for HSS at $P_{24h} > 1\,mm$ were found only for the two sites Christchurch and Kaikoura, both of which are located along the east coast of the South Island of New Zealand. Since

ICAR is based on linear mountain wave theory this result is expected, since improvements for $P_{24h}$ are mainly deemed to manifest themselves in complex topography. In the following, only stations in complex topography are considered.





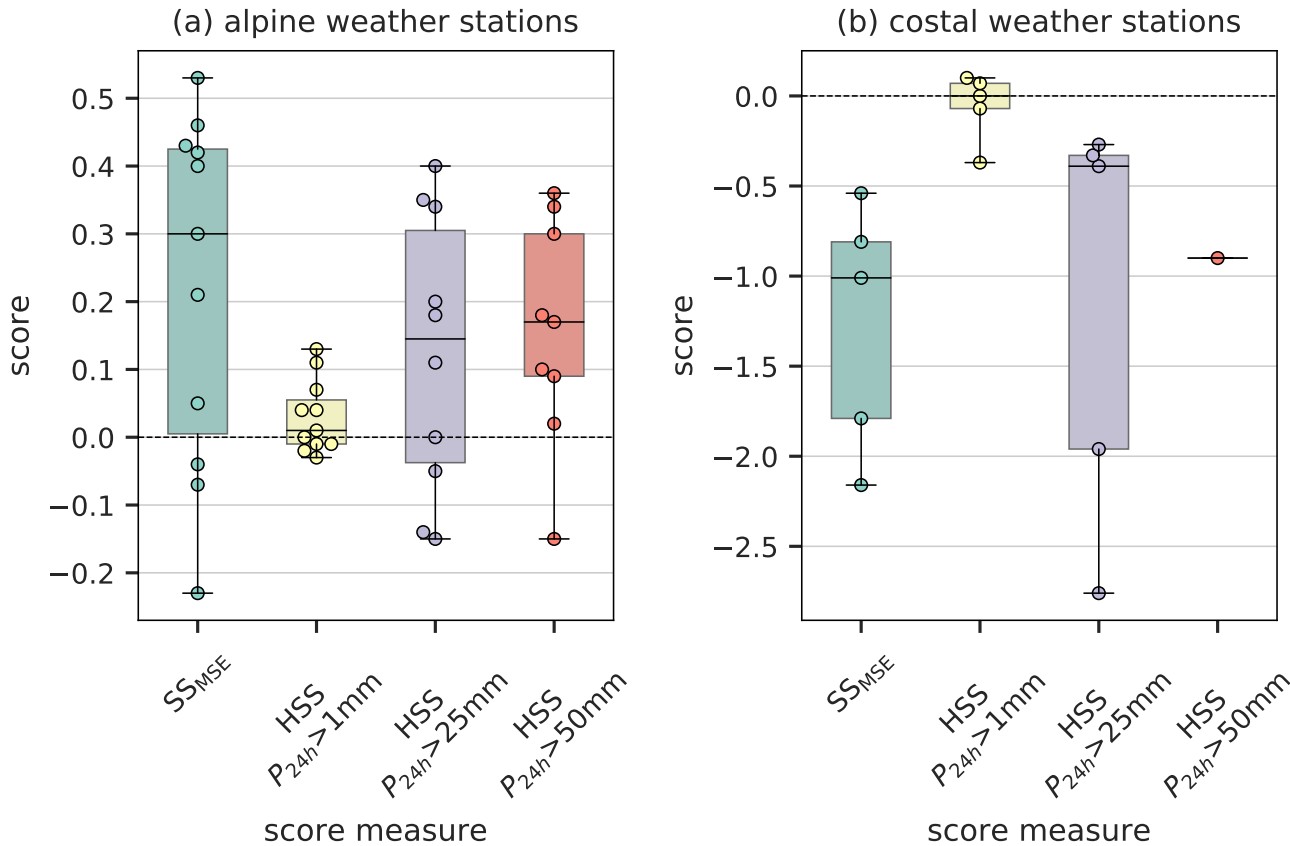

**Figure 3.** Box and whisker plots of all assessed skill scores (x-axis) obtained for ICAR$_{CP}$ with ERAI as reference. All skill scores were calculated using the entire $P_{24h}$ time series available at each weather station for (a) alpine weather stations and (b) coastal weather stations. The lower boundary of the box indicates the 25th percentile, the upper boundary the 75th percentile and the horizontal line the median. Whiskers show the minimum and maximum values of the data set. The circles show the individual values of each skill measure for all stations.




**Table 2.** Time series characteristics for all the weather stations as well as a detailed overview of performance metrics for both ICAR$_{CP}$ and ERAI obtained for each individual site. Empty cells indicate that less than ten days were available for the calculation of the corresponding score. An asteriks (*) preceding a positive score denotes that the score is not significant with regards to the criteria laid out in Section 4.2.

| No | Name | length (yr) | days with $P_{24h}$ above (%) | | | SS$_{MSE}$ (1) | RMSE (mm) | | bias (mm) | | HSS (1) | | |
|----|------|-------------|------|------|------|------|---------|------|---------|------|------|-------|-------|
| | | | 1mm | 25mm | 50mm | | ICAR$_{CP}$ | ERAI | ICAR$_{CP}$ | ERAI | 1mm | 25mm | 50mm |
| Coastal Stations | | | | | | | | | | | | | |
| 1 | Invercargill | 11.0 | 42 | 0.8 | 0.0 | -2.16 | 5 | 3 | 1.9 | 0.2 | -0 | -2.76 | - |
| 2 | Wellington | 11.0 | 29 | 1.3 | 0.2 | -0.54 | 5 | 4 | 0.7 | -0.0 | -0.37 | -0.27 | - |
| 3 | Christchurch | 11.0 | 21 | 0.7 | 0.0 | -1.01 | 4 | 3 | 0.6 | 0.5 | 0.1 | -0.39 | - |
| 4 | Hokitika | 4.7 | 46 | 10.3 | 2.7 | -0.81 | 12 | 9 | 1.0 | -3.2 | -0.07 | -0.33 | -0.9 |
| 5 | Kaikoura | 5.8 | 23 | 1.2 | 0.3 | -1.79 | 8 | 4 | 1.6 | 0.2 | 0.07 | -1.96 | - |
| Alpine Stations | | | | | | | | | | | | | |
| 6 | Arthur Pass | 6.5 | 43 | 16.1 | 7.8 | 0.42 | 18 | 24 | -5.8 | -8.6 | *0.04 | 0.34 | 0.17 |
| 7 | Aoraki/Cook | 10.5 | 41 | 14.0 | 6.1 | 0.46 | 17 | 23 | -3.4 | -6.8 | 0.07 | 0.35 | 0.34 |
| 8 | Albert Burn | 3.2 | 49 | 11.2 | 3.0 | -0.23 | 10 | 9 | -3.6 | -2.5 | -0.03 | -0.05 | 0.09 |
| 9 | Ivory | 6.4 | 53 | 20.4 | 13.2 | 0.53 | 30 | 44 | -5.0 | -16.4 | -0.01 | 0.4 | 0.36 |
| 10 | Brewster | 5.3 | 45 | 25.0 | 11.9 | 0.21 | 22 | 24 | -10.4 | -11.9 | -0.01 | 0.18 | 0.1 |
| 11 | Philistine | 5.4 | 52 | 14.3 | 7.6 | 0.43 | 21 | 28 | -1.0 | -8.8 | -0.02 | 0.2 | 0.18 |
| 12 | Raikai | 6.7 | 44 | 6.8 | 2.2 | *0.05 | 9 | 10 | 0.4 | -1.9 | *0.01 | -0.14 | *0.02 |
| 13 | Mueller Hut | 3.2 | 51 | 14.1 | 7.6 | 0.4 | 25 | 32 | -4.0 | -9.4 | 0 | *0.11 | 0.3 |
| 14 | Larkins | 0.8 | 37 | 2.3 | 0.3 | *0.3 | 5 | 6 | -0.2 | 2.3 | 0.13 | - | - |
| 15 | Mahanga | 5.3 | 43 | 7.2 | 2.7 | -0.04 | 13 | 13 | -1.4 | -2.9 | *0.04 | -0.15 | -0.15 |
| 16 | Potts | 0.9 | 39 | 5.9 | 2.1 | -0.07 | 15 | 15 | -3.0 | -1.0 | *0.11 | 0 | - |





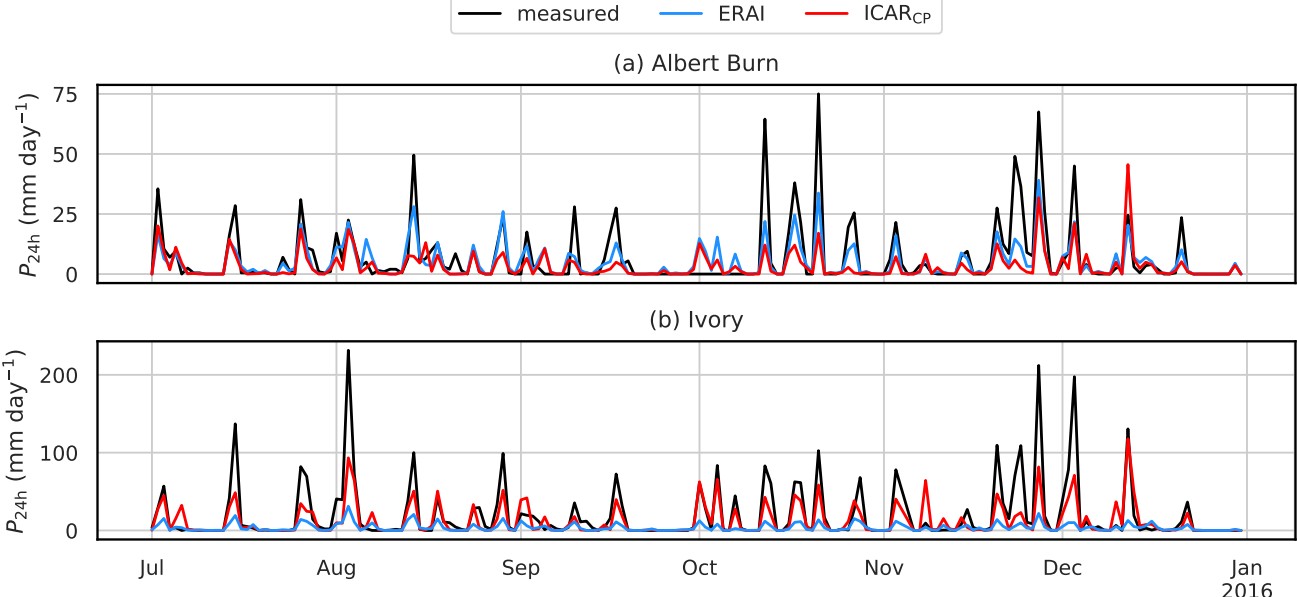

**Figure 4.** Observed and simulated example time series of $P_{24h}$ during the second half of 2015 at (a) Albert Burn and (b) Ivory. At these sites the lowest and highest $SS_{MSE}$ were achieved, during 2015, $SS_{MSE}$ is $-0.39$ for Albert Burn and $0.58$ for Ivory. Both sites are $210\,km$ apart and located at elevations of $120\,m\,MSL$ and $1390\,m\,MSL$ respectively. While Albert Burn lies approximately $11\,km$ downstream of the main alpine ridge, Ivory is located $1\,km$ upstream relative to the predominant westerlies and northwesterlies.

## 4.5 Seasonal Variations of ICAR Performance

Simulations with ICAR show the seasonal variation of precipitation across the South Island. Figure 5 illustrates the 10 year
5  mean daily precipitation $\overline{P_{24h}}$ and seasonal differences to it as computed with four different methods: The VCSR, ICAR, ICAR$_{CP}$ and ERAI. For the weather station data in this study skill measures were calculated for each season individually and are shown in Fig. 6.

     Overall, the average precipitation pattern of VCSR (Fig. 5a) is best captured by ICAR$_{CP}$ (Fig. 5k). While ICAR and ICAR$_{CP}$ patterns are very similar, the former is, when compared to VCSR, too dry to the east of the Southern Alps, particularly between
10  approximately $44°\,S$ and $45°\,S$. However, VCSR indicates larger amounts of precipitation, along the south-west and west coast of the South Island, which are underestimated by ICAR and ICAR$_{CP}$. Furthermore VCSR shows a precipitation maximum in the Southern Alps between $43–44°\,S$ with approximately $20–40\,mm\,day^{-1}$. While this maximum is found in ICAR and ICAR$_{CP}$ patterns, it is confined to a smaller area and shifted westward, located along the $1000\,m\,MSL$ contour line in Fig. 5f and Fig. 5k. Nonetheless, the characteristics of the west-east precipitation profile observed on the South Island of New Zealand (e.g.





Henderson and Thompson, 1999) are captured by ICAR and ICAR$_{CP}$. This is, to some extent, also the case for ERAI (Fig. 5p), albeit with much lower maxima and flatter west-east gradients.

The seasonal variations of precipitation as derived from the VCSR data set (Fig. 5b-e) are best reproduced by ICAR$_{CP}$ (Fig.
5l-o). However, the improvements over the corresponding ICAR patterns 5g-j) are small and the remainder of this paragraph applies to ICAR and ICAR$_{CP}$ alike. When comparing VCSR and ICAR$_{CP}$ the similarities are largest for winter (Fig. 5h and 5m) and summer (Fig. 5e and 5o). The differences increase for the remaining seasons, with the Southern Alps being particularly affected. For autumn, VCSR shows the precipitation as below average (Fig. 5b) while ICAR$_{CP}$ indicates above average precipitation (Fig. 5l). For spring, on the other hand, VCSR shows an increase in precipitation throughout the Southern Alps
(Fig. 5d) but ICAR$_{CP}$ shows the central part of the Southern Alps as drier than on average (Fig. 5n). ERAI, in comparison to VCSR, lacks the fine grid spacing needed to resolve local effects of the topography. However, the patterns roughly capture the seasonal variations of precipitation across the South Island although at a much lower magnitude (Fig. 5q-t).

Seasonal averages of daily accumulated precipitation $\overline{P_{24h}}(\text{se})$ derived from measurements at the alpine weather stations show winter as the driest season, summer as the wettest and the transitional seasons in between (not shown). $\overline{P_{24h}}(\text{se})$ values
as simulated by ICAR$_{CP}$ also correctly show winter as the driest season, autumn in between and summer as the wettest season, with spring as wet as summer in ICAR$_{CP}$. However, $\overline{P_{24h}}(\text{se})$ values derived from ICAR$_{CP}$ underestimate seasonal averages derived from measurements by up to $37\%$ (not shown). ERAI on the other hand is not able to reproduce this pattern in the seasonal averages derived from measurements at all. Here, spring is the wettest season and autumn the driest.

Added value of ICAR$_{CP}$ in terms of SS$_{MSE}$ is found for spring, summer and autumn with median values greater than $0.36$.
For a model based on linear theory, a better performance may be expected during the winter half of the year, when convective available potential energy is lower and convective events are rarer. This is not reflected in the median of SS$_{MSE}$ for winter, which is the lowest of all seasons with $0.08$ and has the largest spread of values (see Fig. 6a). However, the seasonal variation of the mean squared error (RMSE) for ICAR$_{CP}$ shows a minimum during the winter season, see Fig. 6b. This, nonetheless, is the case for ERAI as well, resulting in the lowest RMSEs of ERAI during winter compared to the other seasons. Since the
RMSE decrease during winter is larger for ERAI than it is for ICAR$_{CP}$, this results in a correspondingly lower value of SS$_{MSE}$ in comparison to the other seasons. For HSS the $1\text{mm}$ threshold shows almost no seasonal variation with low median scores of less than $0.05$ during all seasons. At the higher thresholds the pattern is different. For $P_{24h} > 25\text{mm}$ the highest scores are found during autumn and summer with the lowest scores during the remaining seasons. At $P_{24h} > 50\text{mm}$ the seasonal variation is stronger and shows less spread among the stations, with the highest median score during winter and summer and
the lowest scores during the transitional seasons. While ICAR most consistently provides added value at higher thresholds, site specific improvements are observed even at $P_{24h} > 1\text{mm}$.





**Figure 5.** The top four panels show patterns of $P_{24h}$ averaged over 2007–2016 for VCSR (left), ICAR (second column), ICAR$_{CP}$ (third column) and ERAI (right) over the South Island of New Zealand and surrounding ocean. Rows two to five show seasonal deviations of the all-year average patterns, for autumn (second row), winter (third row), spring (fourth row) and summer (bottom). Each panel shows the coastline and the $1000\,\text{m}$ MSL contour line of the topography.





**Figure 6.** Panel (a) shows values of $SS_{MSE}$ and HSS (from left to right) for all seasons (colors of the boxes) and panel (b) the root mean squared errors RMSE of $ICAR_{CP}$ and ERAI for all alpine stations. Each box and whisker plot is associated with a season, indicated by box color, and a skill measure (x-axis). The lower boundary of each boxplot indicates the 25th percentile, its upper boundary the 75th percentile and a black line the median. Whiskers show the minimum and maximum values of the data set. Circles on top of the boxes show the individual values of each skill measure for all stations.



## 4.6 Sensitivity of ICAR performance to upstream flow linearity

As a model that is based on linear theory, ICAR is expected to perform best in cases where linear theory is a valid approximation of the atmospheric flow at the sites of interest. An indicator of whether this is the case or not is the non-dimensional mountain height (e.g. Smith, 1980), from here on referred to as Froude number $F$:

$$F = \frac{U_n}{NH}. \tag{4}$$

Here $U_n$ denotes the horizontal wind speed perpendicular to the Southern Alps, $N$ the Brunt-Väisälä frequency and $H$ an assumed homogenous ridge height of $1500\,\mathrm{m}$ characterizing the Southern Alps. Values of $F$ equal or larger than unity indicate linear flow, while values of $F$ closer to zero point towards non-linearity (Smith, 1980).

In order to derive $U_n$ and $N$, two volumes upstream of the west and east coast were defined, from which the properties of the flow at an angle of $90 \pm 20°$ to the Southern Alps were extracted from ERAI daily averages. They are located $200\,\mathrm{km}$ northwest and southeast of the west and east coast of the South Island, respectively to minimize the effect of the ERAI topography on the flow. Each volume is oriented parallel to the corresponding coast and is about $200\,\mathrm{km}$ wide, $1000\,\mathrm{km}$ long and $1500\,\mathrm{m}$ high, each containing 22 ERAI grid points. For northwesterly flow, properties were extracted from the volume to the northwest of the western coast, and for southeasterlies from the volume southeast of the eastern coast.

Following the approach of Reinecke and Durran (2008), the Brunt-Väisälä frequency and wind speed perpendicular to the Southern Alps were calculated with the averaging method for each ERAI gridpoint in the volumes:

$$\overline{N} = \frac{1}{H} \int_0^H N(z)\,dz \tag{5}$$

$$\overline{U_n} = \frac{1}{H} \int_0^H U_n(z)\,dz, \tag{6}$$

where $\overline{N}$ and $\overline{U_n}$ are the averages of the Brunt-Väisälä frequency and wind speed perpendicular to the Southern Alps respectively, weighted by the thickness of the vertical levels. For a relative humidity RH below $90\%$ the dry Brunt-Väisälä frequency was employed in equation (5), while for RH larger or equal to $90\%$ the moist Brunt-Väisälä frequency $N_m$ (Emanuel, 1994) was used:

$$N^2 = g\frac{d\ln\theta}{dz}, \tag{7}$$

$$N_m^2 = \frac{1}{1+q_w}\left\{\Gamma_m\frac{d}{dz}\left[(c_p + c_l q_w)\ln\theta_e\right] - \left[c_l\Gamma_m\ln T + g\right]\frac{dq_w}{dz}\right\}. \tag{8}$$

Here $g$ is the acceleration due to gravity, $T$ the temperature, $\theta$ the potential temperature, $\theta_e$ the equivalent potential temperature, $\Gamma_m$ the saturated adiabatic lapse rate, $c_p$ and $c_l$ the specific heats at constant pressure of dry air and liquid water, $q_s$ the saturation mixing ratio, $q_l$ the liquid water mixing ratio and the total water content $q_w = q_s + q_l$.

$F$ was then calculated from the weighted averages of $\overline{N}$ and $\overline{U_n}$ at all grid points showing stable atmospheric conditions. The imaginary part of the weighted average of the Brunt-Väisälä frequency $\overline{N}_i$, was used as an indicator of whether the atmosphere





at an ERAI grid point was stably stratified. For $\overline{N}_i$ below a threshold $\kappa$ the stratification was considered stable, while $\overline{N}_i$ larger or equal to $\kappa$ was classified as near-stable. The nomenclature "near-stable" is chosen over "unstable" since vertical potential temperature profiles indicated that the nonzero imaginary part of $\overline{N}_i$ in the large majority of cases is caused by a

thin unstable layer close to the ocean surface, not representative of the conditions above and with a negligible effect on flow linearity. To investigate the dependence of the results on the threshold choice, the value of $\kappa$ is varied between $25 \cdot 10^{-5}\,\mathrm{s}^{-1}$ and $600 \cdot 10^{-5}\,\mathrm{s}^{-1}$ in steps of $25 \cdot 10^{-5}\,\mathrm{s}^{-1}$. If more than half of the grid points in an upstream volume showed near-stable conditions, flow for this day was classified accordingly. Otherwise the day was marked as having stable atmospheric flow with an average Froude number $\overline{F}$. Days when the volume to the northwest and the volume to the southeast both showed flow

towards the Southern Alps were excluded from the analysis. This procedure allowed to categorize all remaining days in the eleven-year study period into days when atmospheric conditions upstream of the weather stations were either (i) near-stable ($\overline{N}_i \geq \kappa$), (ii) stable with flow of low linearity ($\overline{F} < 1$) or (iii) stable with flow of high linearity ($\overline{F} \geq 1$). All data from alpine weather stations were then grouped by these categories and skill scores calculated to analyze ICAR performance with regard to the atmospheric background state.

Of the $4018$ days in the eleven-year study period, $3887$ fulfill criteria (i) to (iii) specified above and depending on the choice of $\kappa$ a different number of days is assigned to each category. A detailed overview over the distribution of days among the three categories in dependence of $\kappa$ is given in Table 3. The results from Table 3 summarized in Fig. 7 show, that stable atmospheric conditions and Froude numbers larger or equal to unity lead to an increase in median scores for sites in complex topography. This behavior is observed for $\mathrm{SS}_{\mathrm{MSE}}$ where the score median increases from $0.34$ to $0.50$ and, for $P_{24h} > 25\,\mathrm{mm}$

and $P_{24h} > 50\,\mathrm{mm}$ in case of HSS. For $P_{24h} > 1\,\mathrm{mm}$ the trend is less pronounced and depends much more on the choice of $\kappa$. The score maximum in Fig. 7b of $0.5$ for stable atmospheric conditions with $\overline{F} \geq 1$ was found to be non-significant (see Table 3) according to the criteria in Sect. 4.2. The spread of scores indicated by the whiskers in Fig. 7 is discussed separately in Sect. 5. Notably the analysis shows that $\mathrm{ICAR}_{\mathrm{CP}}$ not only provides added value over ERAI during stable days with high flow linearity, but during near-stable days and stable days with low flow linearity as well.



**Table 3.** Skill measures calculated for the three categories of atmospheric flow (near-stable, stable with $\overline{F} < 1$ and stable with $\overline{F} \geq 1$) and number of days pertaining to each category in percent in dependence of $\kappa$. An asterisk preceding a score indicates that it was found to be non-significant by applying the criteria defined in Sect. 4.2.

| $\kappa\ (10^{-5}\,\mathrm{s}^{-1})$ | 25 | 50 | 75 | 100 | 125 | 150 | 175 | 200 | 275 | 300 | 375 | 400 | 425 | 500 | 525 | 575 | 600 |
|---|---|---|---|---|---|---|---|---|---|---|---|---|---|---|---|---|---|
| days in each category (%) | | | | | | | | | | | | | | | | | |
| days near-stable | 75.6 | 57.8 | 41.6 | 29.3 | 19.5 | 13.1 | 9.7 | 7.2 | 2.9 | 2.4 | 1.1 | 0.9 | 0.7 | 0.3 | 0.2 | 0.1 | 0.1 |
| days stable, $\overline{F} < 1$ | 24.2 | 41.7 | 57.7 | 69.8 | 79.3 | 85.4 | 88.7 | 91 | 95 | 95.5 | 96.8 | 97 | 97.2 | 97.5 | 97.7 | 97.8 | 97.8 |
| days stable, $\overline{F} \geq 1$ | 0.2 | 0.5 | 0.6 | 0.9 | 1.2 | 1.4 | 1.6 | 1.8 | 2.1 | 2.1 | 2.1 | 2.1 | 2.1 | 2.1 | 2.1 | 2.1 | 2.1 |
| scores for near-stable days | | | | | | | | | | | | | | | | | |
| $SS_{MSE}$ | 0.4 | 0.38 | 0.33 | 0.35 | 0.35 | 0.36 | 0.41 | 0.44 | 0.54 | 0.58 | *0.18 | *0.2 | -0.21 | -0.22 | -0.54 | -0.11 | -0.13 |
| HSS $P_{24h} > 1\,\mathrm{mm}$ | 0.03 | 0.03 | 0.03 | *0.01 | *0.02 | 0 | 0 | 0 | *0.01 | *0.02 | *0.02 | -0.05 | -0.06 | -0.36 | -0.22 | 0 | 0 |
| HSS $P_{24h} > 25\,\mathrm{mm}$ | 0.2 | 0.15 | 0.1 | 0.07 | -0.02 | -0.08 | -0.07 | -0.1 | -0.12 | 0 | -0.36 | -0.38 | -0.13 | -0.14 | -2 | - | - |
| HSS $P_{24h} > 50\,\mathrm{mm}$ | 0.18 | 0.16 | 0.11 | 0.09 | *0.08 | *0.07 | *0.09 | *0.12 | *0.18 | *0.25 | *0.22 | *0.22 | 0 | 0 | - | - | - |
| scores for stable days with $\overline{F} < 1$ | | | | | | | | | | | | | | | | | |
| $SS_{MSE}$ | 0.46 | 0.44 | 0.45 | 0.43 | 0.42 | 0.42 | 0.41 | 0.41 | 0.4 | 0.4 | 0.41 | 0.41 | 0.41 | 0.41 | 0.41 | 0.41 | 0.41 |
| HSS $P_{24h} > 1\,\mathrm{mm}$ | -0.01 | *0.01 | *0.02 | 0.03 | *0.03 | 0.03 | 0.03 | 0.03 | 0.03 | 0.03 | *0.02 | 0.02 | *0.02 | 0.03 | 0.02 | 0.02 | 0.02 |
| HSS $P_{24h} > 25\,\mathrm{mm}$ | 0.3 | 0.31 | 0.3 | 0.28 | 0.28 | 0.26 | 0.25 | 0.25 | 0.23 | 0.23 | 0.22 | 0.22 | 0.22 | 0.22 | 0.22 | 0.22 | 0.22 |
| HSS $P_{24h} > 50\,\mathrm{mm}$ | 0.28 | 0.26 | 0.26 | 0.24 | 0.23 | 0.23 | 0.23 | 0.22 | 0.21 | 0.21 | 0.21 | 0.21 | 0.21 | 0.21 | 0.21 | 0.21 | 0.21 |
| scores for stable days with $\overline{F} \geq 1$ | | | | | | | | | | | | | | | | | |
| $SS_{MSE}$ | 0.48 | 0.59 | 0.63 | 0.58 | 0.52 | 0.51 | 0.48 | 0.48 | 0.48 | 0.48 | 0.5 | 0.5 | 0.5 | 0.5 | 0.5 | 0.5 | 0.5 |
| HSS $P_{24h} > 1\,\mathrm{mm}$ | *0.5 | 0 | 0 | *0.1 | 0 | 0 | 0 | *0.05 | *0.04 | *0.04 | *0.04 | *0.04 | *0.04 | *0.04 | *0.04 | *0.04 | *0.04 |
| HSS $P_{24h} > 25\,\mathrm{mm}$ | *0.27 | 0.37 | 0.45 | 0.49 | 0.5 | 0.44 | 0.42 | 0.39 | 0.36 | 0.36 | 0.38 | 0.38 | 0.38 | 0.38 | 0.38 | 0.38 | 0.38 |
| HSS $P_{24h} > 50\,\mathrm{mm}$ | 0.46 | 0.49 | 0.57 | 0.44 | 0.41 | 0.36 | 0.3 | 0.29 | 0.28 | 0.28 | 0.31 | 0.31 | 0.31 | 0.31 | 0.31 | 0.31 | 0.31 |
| near-stable days used for HSS score calculation (%) | | | | | | | | | | | | | | | | | |
| with $P_{24h} > 1\,\mathrm{mm}$ | 34.7 | 26.7 | 20.1 | 15.3 | 10.8 | 7.6 | 5.9 | 4.4 | 1.8 | 1.4 | 0.6 | 0.6 | 0.4 | 0.3 | 0.1 | 0.1 | 0.1 |
| with $P_{24h} > 25\,\mathrm{mm}$ | 10 | 7.3 | 5.3 | 3.9 | 2.7 | 1.9 | 1.5 | 1.1 | 0.5 | 0.4 | 0.1 | 0.1 | <0.1 | <0.1 | <0.1 | 0 | 0 |
| with $P_{24h} > 50\,\mathrm{mm}$ | 4.6 | 3.2 | 2.3 | 1.6 | 1 | 0.7 | 0.6 | 0.5 | 0.2 | 0.2 | <0.1 | <0.1 | <0.1 | <0.1 | 0 | 0 | 0 |
| stable days with $\overline{F} < 1$ used for HSS score calculation (%) | | | | | | | | | | | | | | | | | |
| with $P_{24h} > 1\,\mathrm{mm}$ | 12.3 | 20.3 | 27.2 | 31.8 | 35.8 | 38.6 | 40 | 41.2 | 43.6 | 43.9 | 44.8 | 44.9 | 45 | 45.2 | 45.2 | 45.3 | 45.3 |
| with $P_{24h} > 25\,\mathrm{mm}$ | 4.5 | 7.2 | 9.2 | 10.6 | 11.5 | 12.1 | 12.3 | 12.6 | 13.1 | 13.1 | 13.3 | 13.4 | 13.4 | 13.4 | 13.4 | 13.4 | 13.4 |
| with $P_{24h} > 50\,\mathrm{mm}$ | 2.3 | 3.7 | 4.6 | 5.2 | 5.6 | 5.8 | 5.8 | 5.9 | 6.1 | 6.1 | 6.2 | 6.2 | 6.2 | 6.2 | 6.2 | 6.2 | 6.2 |
| stable days with $\overline{F} \geq 1$ used for HSS score calculation (%) | | | | | | | | | | | | | | | | | |
| with $P_{24h} > 1\,\mathrm{mm}$ | 0.1 | 0.4 | 0.5 | 0.7 | 1 | 1.2 | 1.3 | 1.4 | 1.6 | 1.6 | 1.6 | 1.6 | 1.7 | 1.6 | 1.6 | 1.6 | 1.6 |
| with $P_{24h} > 25\,\mathrm{mm}$ | 0.1 | 0.3 | 0.3 | 0.4 | 0.7 | 0.8 | 0.9 | 1 | 1.1 | 1.1 | 1.1 | 1.1 | 1.1 | 1.1 | 1.1 | 1.1 | 1.1 |
| with $P_{24h} > 50\,\mathrm{mm}$ | 0.1 | 0.2 | 0.3 | 0.4 | 0.5 | 0.6 | 0.7 | 0.7 | 0.8 | 0.8 | 0.8 | 0.8 | 0.8 | 0.8 | 0.8 | 0.8 | 0.8 |





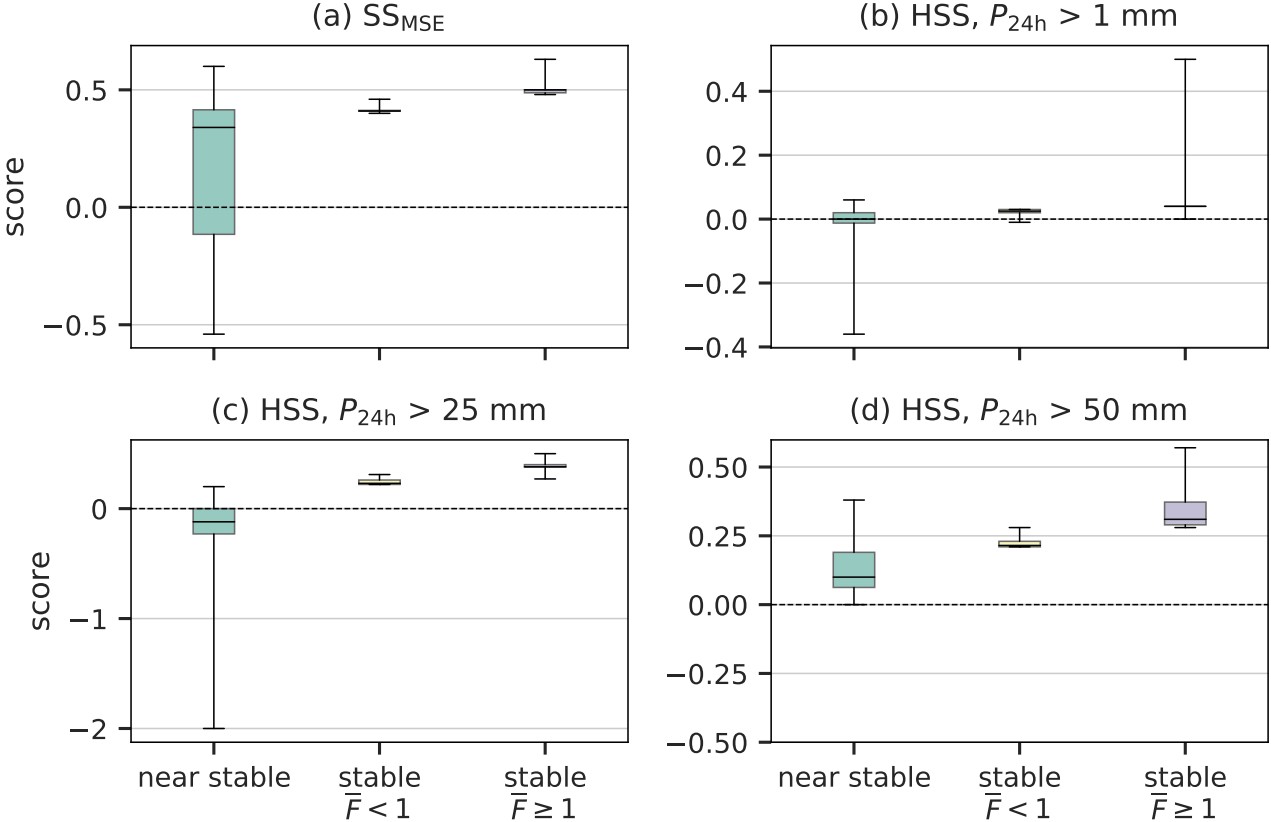

**Figure 7.** Dependence of SS$_{MSE}$ and HSS at alpine stations on atmospheric stability and the Froude number regime, calculated for all available data for each value of $\kappa$ (see Table 3). SS$_{MSE}$ is shown in (a) and HSS for thresholds (b) $P_{24h} > 1$mm, (c) $P_{24h} > 25$mm and (d) $P_{24h} > 50$mm. The x-axis indicates atmospheric stability and Froude number regime. The lower boundary of each boxplot indicates the 25th percentile, its upper boundary the 75th percentile and a black line the median. Whiskers show the minimum and maximum values of the data set.

## 4.7 Weather Pattern Based Evaluation of ICAR

Kidson (1994a) developed a daily weather pattern classification scheme for New Zealand based on 24 h mean sea-level pressure
5  fields. For the underlying cluster analysis the NCEP/NCAR 40-year reanalysis dataset (Kalnay et al., 1996) between January 1958 and June 1997 was employed. This analysis yielded twelve synoptic weather patterns (Kidson, 2000) associated with three regimes: Trough, Zonal and Blocking. The Trough regime is characterized by troughs crossing New Zealand and above average precipitation countrywide; the Zonal regime by strong zonal flow to the south and highs to the north with milder conditions in the south; and the Blocking regime by highs in the south leading to a dryer southwest but wetter northeast. On





average about 38% of days are classified as belonging to the Trough regime, 25% to the Zonal regime and 37% to the Blocking regime. Figure 8 gives an overview of the twelve synoptic weather patterns defined for New Zealand and the associated regime. An operational pattern-classification of each day since 1948 is available from the National Institute of Water and Atmospheric Research of New Zealand (NIWA).







**Figure 8.** Synoptic weather patterns and their associated regimes for New Zealand. Each panel lists the pattern identifier and its relative frequency in brackets next to it, while the contour lines depict the geopotential height (m) at 1000hPa. Reproduced from Kidson (2000), copyright Royal Meteorological Society.





Furthermore, these weather patterns have been linked to deviations of quantities, such as precipitation, from the climatological mean (Kidson, 1994b, 2000). For instance, during the HW pattern precipitation is below average at all weather stations, while during the TNW, T, HE and W patterns, when westerlies and northwesterlies dominate and orographic lifting in the Southern
5  Alps is favored, precipitation at all alpine weather stations is above average, see, for example, Sect. 4.8. This allows for the investigation of whether a downscaling model is able to represent these departures correctly, offering a link between the synoptic situation and local weather anomalies.

    Figure 9 shows a distinct dependence of $SS_{MSE}$ on the synoptic weather pattern. Highest median scores with values above 0.29 in terms of $SS_{MSE}$ are achieved for the weather patterns TNW, T, H and W. Three of these patterns (TNW, T and W)
10  are associated with distinct westerly and northwesterly flow, facilitating orographic lifting along the Southern Alps. However, the HE pattern, for which similar conditions may be expected, only yields a median $SS_{MSE}$ of 0.15. This comparatively small median value is due to very low scores found for the three weather stations Potts, Raikai and Mahanga. Raikai and Potts are farthest downwind of the main alpine ridge and Mahanga is the weather station farthest downwind of the coast with approximately $80\mathrm{km}$ of mountainous terrain in between, where downwind is as defined in Sect. 3.2. Particularly low median
15  scores are found for the patterns HW and NE, where flow parallel to the Southern Alps dominates. Consistent with the results from Sect. 3, no added value of $ICAR_{CP}$ over ERAI was found in terms of HSS for $P_{24h} > 1\mathrm{mm}$, even though there is a small variation with weather pattern (not shown). For the higher thresholds not enough data were available to calculate HSS for every weather pattern.

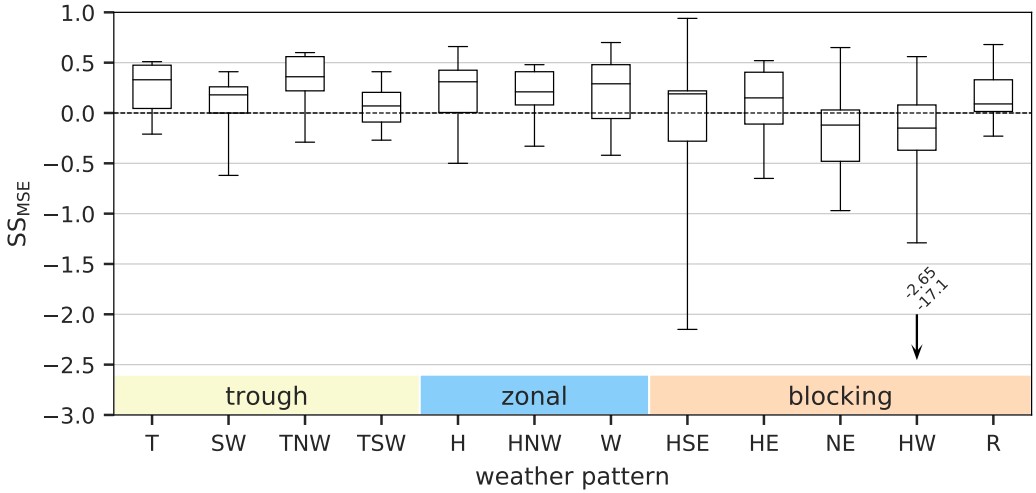

**Figure 9.** Box and whisker plot of $SS_{MSE}$ calculated for all alpine weather station in dependence of the synoptic weather pattern (Kidson, 2000). The color shadings in the lower part of the plot indicate the regime, the weather patterns listed on the x-axis are associated with. The lower bound of each box marks the 25th percentile of the data, the upper bound the 75th and the black horizontal line the median. The whiskers indicate minimum and maximum values in the data set, except for the HW pattern where two data points outside the plot limit are indicated by an arrow and their corresponding values written above.





## 4.8 Weather pattern based variations of precipitation

It has been noted by Kidson (1994b) that the local climate in New Zealand shows variability in dependence of the synoptic weather patterns. In this section, the capability of ICAR to capture the average 24h accumulated precipitation amount at each weather station (ws) calculated for each of the weather patterns (wp) is investigated. To this end averages of $P_{24h}$ simulated by ICAR and ERAI are calculated individually for each weather pattern and each of the weather stations $\overline{P_{24h}}(\text{ws}, \text{wp})$ and compared to the observations.

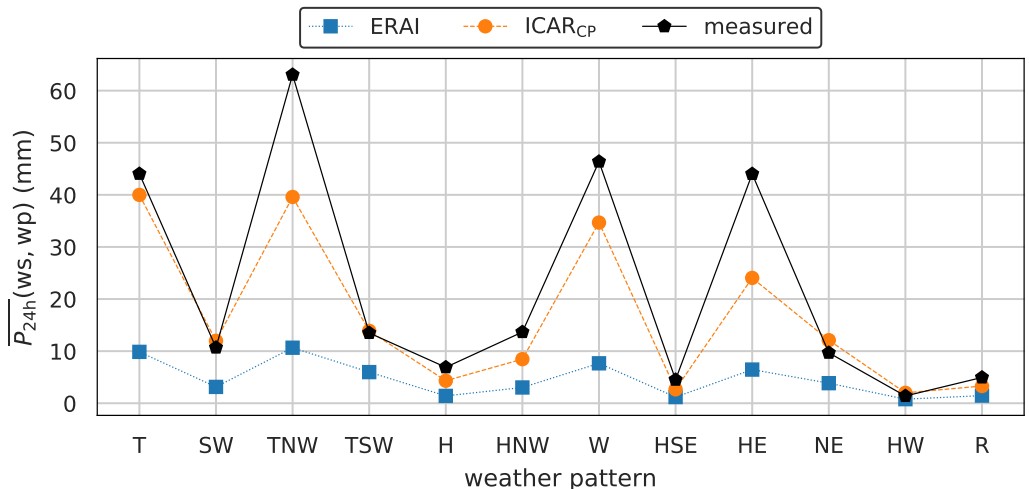

**Figure 10.** $\overline{P_{24h}}(\text{ws}, \text{wp})$ as a function of weather pattern (wp) and weather station (ws) at weather station Ivory for measurements (black pentagons), ICAR simulations (orange disks) and the ERAI reanalysis (blue squares). Ivory is situated at an elevation of $1390\,\text{m MSL}$ and, on average, approximately $2\,\text{km}$ upstream of the main alpine ridge with regard to northwesterlies and westerlies. The connecting lines serve as guides to the eye.

Figure 10 shows measured and modeled values of $\overline{P_{24h}}(\text{ws}, \text{wp})$ for the weather station Ivory. It is located at an elevation of $1390\,\text{m MSL}$ and lies approximately $2\,\text{km}$ upstream of the main alpine ridge with respect to westerly and northwesterly flow (see Sect. 3.2). Ivory is strongly affected by precipitation caused by orographic lifting, leading to local precipitation maxima during the T, TNW, W and HE patterns. For Ivory the trend found in the measurements is correctly reproduced by $\text{ICAR}_{\text{CP}}$ and ERAI, the absolute amounts of precipitation are, while underestimated, better modeled by $\text{ICAR}_{\text{CP}}$. To analyze how well the simulated values of $\overline{P_{24h}}(\text{ws}, \text{wp})$ correlate with measurements, the coefficient of determination weighted by weather pattern frequency, $r^2$, (Wilks, 2011b, Chapter 5) between the observed and modeled values of $\overline{P_{24h}}(\text{ws}, \text{wp})$ are calculated for all weather stations and shown in Fig. 11a. To investigate the added value of $\text{ICAR}_{\text{CP}}$ over ERAI in modeling measured amounts of $\overline{P_{24h}}(\text{ws}, \text{wp})$, $\text{SS}_{\text{MSE}}$ is calculated and the results are summarized in Fig. 11b.



With the exception of the weather station Potts, $ICAR_{CP}$ is able to represent the fluctuation of $\overline{P_{24h}}(ws,wp)$ as a function of weather pattern, with $r^2$ higher than $0.9$ (see Fig. 11a). $ICAR_{CP}$ shows clear improvement over ERAI at five of eleven weather stations, a similar performance to ERAI at four and a worse performance at two. Particularly noteworthy is the underperformance in comparison to ERAI at the alpine weather station Potts and, far less pronounced, at weather station Larkins. Both are located downstream of the main alpine ridge but only at Potts does $ICAR_{CP}$ not correctly anticipate decreased precipitation during the HW and TNW, as well as an increase in precipitation during the W pattern (not shown).

Generally $ICAR_{CP}$ is able to model measured amounts of $\overline{P_{24h}}(ws,wp)$ well at all other alpine weather stations (see Fig. 11b) with a median $SS_{MSE}$ of $0.74$, except for Albert Burn and Potts. At Albert Burn it underestimates measured and ERAI modeled values of $\overline{P_{24h}}(ws,wp)$ during all patterns (not shown). Albert Burn is located approximately $11km$ downwind of the main alpine ridge with respect to westerlies and northwesterlies. The lowest score is found at the alpine weather station Potts. The performance of individual stations is discussed in Sect. 5.





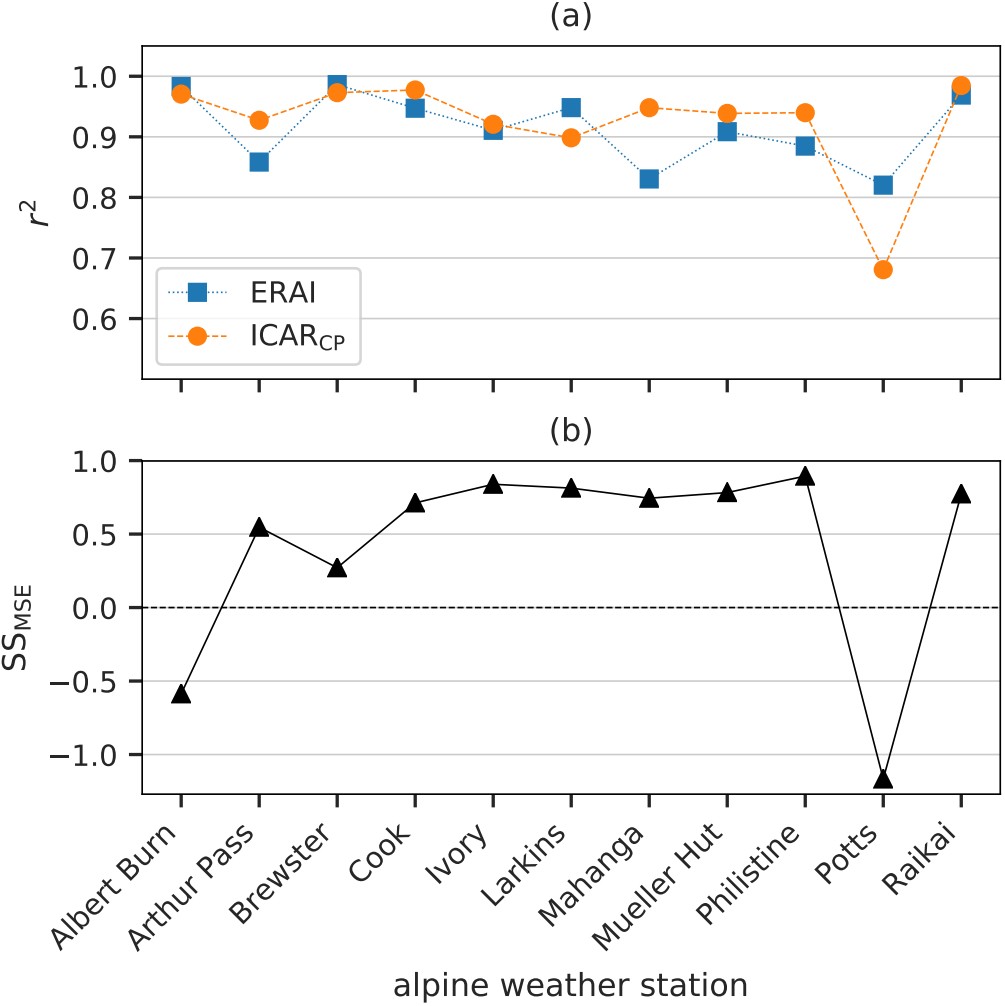

**Figure 11.** Panel (a) shows the coefficient of determination $r^2$ between modeled and measured $\overline{P_{24h}}(\mathrm{ws},\mathrm{wp})$ calculated for each alpine weather station for ICAR$_{CP}$ (orange disks) and ERAI (blue squares). Panel (b) shows the SS$_{MSE}$ of ICAR$_{CP}$ over ERAI in modeling $\overline{P_{24h}}(\mathrm{ws},\mathrm{wp})$ at alpine weather stations. The connecting lines serve as guides to the eye.

## 5 Discussion

The model top leading to the smallest mean MSE of ICAR$_{CP}$ over all alpine weather stations was determined with a sensitivity
5    study at $4\,\mathrm{km}$ above topography. At alpine sites in complex topography ICAR$_{CP}$ is then able to reduce mean squared errors
in comparison to its ERAI forcing dataset by up to $53\,\%$. While ICAR$_{CP}$ models the occurrence of days with a maximum



accumulated precipitation of $1\,\mathrm{mm}$ similarly well as ERAI, significant improvements are found for $P_{24h} > 25\,\mathrm{mm}$ and $P_{24h} > 50\,\mathrm{mm}$. Overall the mean daily precipitation pattern produced by ICAR$_{\mathrm{CP}}$ was found to be in agreement with the pattern derived from the observation-based gridded data set VCSR, with the seasonal variation being mostly captured by ICAR$_{\mathrm{CP}}$. The results

indicate that ICAR$_{\mathrm{CP}}$ performs best during stable atmospheric conditions with flow of high linearity, however, added value over ERAI is found for stable days with low flow linearity and near-stable days as well. A clear dependence of skill on the synoptic situation was found, with weather patterns associated with cross-alpine flow leading to higher scores than weather patterns with flow parallel to the alpine range.

Precipitation measurements and particularly those in complex topography are associated with uncertainties. Different factors

such as wetting, wind, freezing or equipment failure in the harsh conditions (Henderson and Thompson, 1999) may introduce errors, such as undercatch, into the results. Wind has been recognized as the main cause of undercatch (e.g. Groisman and Legates, 1995; Yang et al., 1999; Yang and Ohata, 2001), which affects alpine weather stations in particular. The effect is most pronounced for large, solid precipitation and increases with latitude and elevation (Goodison et al., 1989; Groisman and Legates, 1995). Cullen and Conway (2015), for instance, estimate the undercatch at Mount Brewster during summer with

$25\,\%$, while Kerr et al. (2011) lists annual undercatch at alpine sites in the Southern Alps with up to $20\,\%$. However, the impact of undercatch on the results presented here is expected to be small since these errors have an adverse effect not only on the performance of ICAR$_{\mathrm{CP}}$ but also the ERAI reference model.

The sensitivity studies leading to the choice of the model top at $4\,\mathrm{km}$ have shown that the model top elevation greatly influences precipitation amounts and in turn the obtained mean squared errors, see Fig. 2. It is not immediately obvious though

why precipitation amounts decrease (not shown) and the MSEs deteriorates for higher model tops. Further studies are required to find a method that allows the estimation the model top elevation best suited for a domain without relying on measurements, as well as to investigate the influence of the choice of the forcing data type (global or regional reanalyses, GCMs, weather forecast models) and the spatial grid resolution thereof on ICAR dynamics and skill.

In the analysis presented, standard verification scores based on point matches between model and observation were employed

(see Sect. 4.2). Nonetheless, these verification scores are susceptible to small spatial shifts in the ICAR$_{\mathrm{CP}}$ precipitation field that cannot be produced by the coarse scale reference model. Therefore, this effect may potentially over-penalize ICAR$_{\mathrm{CP}}$ in comparison to the much coarser ERAI field (Theis et al., 2005; Ebert, 2008). An over-penalization of ICAR in comparison to ERAI is suggested by the precipitation pattern comparisons shown Fig. 5. Here the observation-based gridded dataset VCSR and ICAR$_{\mathrm{CP}}$ are generally in good agreement, with ICAR$_{\mathrm{CP}}$ reproducing most seasonal variations. As noted in section 4.5, for

instance, a precipitation maximum in the VCSR pattern (Fig. 5a) that is located within the Southern Alps is shifted westward in the ICAR$_{\mathrm{CP}}$ pattern (Fig. 5k) and is, due the coarser grid-spacing, not present in ERAI at all (and Fig. 5p). A variety of methods have been proposed to overcome this problem and future evaluations of ICAR generated atmospheric fields could incorporate these methods in their evaluation procedures (e.g. Ebert, 2009).

ICAR was found to perform better for upstream flows with Froude numbers larger than unity. This result is not unexpected,

since linear theory is the theoretical foundation for ICAR. Therefore, flows of higher linearity lead to increased SS$_{\mathrm{MSE}}$ and HSS for thresholds of $25\,\mathrm{mm}$ and $50\,\mathrm{mm}$. These results hold even if the method for classifying near-stable or stable days is changed.



For instance, using $\overline{N^2} \leq 0$ as classification criterion for near-stable days and $\overline{N^2} > 0$ for stable days leads to similar results (not shown). However, for $SS_{MSE}$ and HSS at the $50\,\text{mm}$ threshold (see Fig. 7a and Fig. 7d) the spread of scores derived from varying $\kappa$ for near-stable days is large enough to include the median score of the stable days with $\overline{F} \geq 1$. Nonetheless, in only

two ($\kappa = 275 \cdot 10^{-5}\,\text{s}$ and $\kappa = 300 \cdot 10^{-5}\,\text{s}$) out of 17 cases the calculated $SS_{MSE}$ is larger for near-stable days than that for stable days with high flow linearity. For HSS at $P_{24h} > 50\,\text{mm}$ this is never the case. Therefore these two instances are considered as outliers. A potential issue with the methodology is the small number of cases in the stable regime with $\overline{F} \geq 1$ compared to the two other classes (see Table 3). However, $\overline{P_{24h}}$ on stable days with $\overline{F} \geq 1$ is five times as high as $\overline{P_{24h}}$ during the other two classes (not shown). Therefore, while comparably small in number, stable days with $\overline{F} \geq 1$ contribute above-average amounts

of precipitation to the climatology, highlighting the importance of the improvement in skill for this category.

Negative values of $SS_{MSE}$ were found for the alpine weather stations Albert Burn, Mahanga and Potts, while non significant positive scores were found at Raikai and Larkins. The time series of Potts and Larkins are the shortest of all weather stations, spanning $0.9$ years and $0.8$ years respectively, potentially contributing to the negative or non-significant positive score respectively. Potts, additionally, is the weather station with the largest difference between weather station elevation and ICAR

grid cell elevation, with the ICAR grid cell located $741\,\text{m}$ lower. While the aforementioned issues may deteriorate scores at individual stations, it is also possible that the downwind distribution of moisture by ICAR differs from expectations. This is indicated by a slight negative correlation of score value with the average distance downwind from the main alpine crest (as defined in Sect. 3.2) which is found for $SS_{MSE}$ and HSS at the $25\,\text{mm}$ and $50\,\text{mm}$ thresholds. The correlation is strongest for $SS_{MSE}$ with $-0.65$ and weakest for the HSS with $P_{24h} > 50\,\text{mm}$ with $-0.50$. Mahanga and Larkins are the weather stations

farthest downwind from the coast, with mountainous topography in between. Albert Burn, Raikai and Potts are the weather stations farthest downwind of the main alpine crest. A potential cause for the observed negative correlation is, that the reflection of mountain waves at the interfaces between atmospheric layers can have a significant impact on the amount and distribution of precipitation (Siler and Durran, 2015). The implementation of mountain wave reflections in ICAR could therefore lead to improvements in this regard.

The mean $SS_{MSE}$ of $ICAR_{CP}$ at alpine weather stations is $0.3$. While $ICAR_{CP}$ provides added value over ERAI it nonetheless systematically underestimates precipitation at all alpine weather stations except for Raikai (see Table 1 and Table 2). This underestimation increases with higher model top settings and is independent of the average distance of the site up- or downwind of the main alpine ridge (with respect to northwesterlies and westerlies). Different issues may contribute to this underestimation: (i) Potentially ERAI is too dry in the study region and therefore not enough moisture is advected across the boundary of

the nested ICAR domain. (ii) Since the coupling between surface and atmosphere is neglected in the ICAR setup employed for this study, parts of the ocean within the ICAR domain cannot contribute moisture to the airflow upwind of the South Island of New Zealand. (iii) Non-linear amplification of waves could amplify updrafts in comparison to updrafts predicted by linear theory, increasing orographic precipitation correspondingly. (iv) The low model top setting at $4\,\text{km}$ above topography, determined as optimal by a sensitivity study, may largely eliminate potential seeder feeder interaction between synoptic clouds and

orographically lifted moist air. This effect is expected to play a crucial role for the formation of heavy rainfall on the South Island of New Zealand (Purdy and Austin, 2003). (v) Convergences and divergences in the ERAI data set influence updrafts





and downdrafts in the ICAR wind field, leading, for instance, to synoptic precipitation in ICAR. However, these divergences may also dampen the updrafts calculated with linear theory, thereby reducing the precipitation computed by ICAR. (vi) The reflection of mountain waves is neglected by the version of ICAR used in this study. However, Siler and Durran (2015) have

shown that the reflection of mountain waves has a significant impact on the amount and distribution of precipitation. Further studies are needed to quantify the influence of issues (i)–(vi) and identify their relevance for the observed underestimation. A possible ad hoc solution to the underestimation is the application of a bias-correction field estimated from a regional climate model to the ICAR precipitation fields (e.g. Engelhardt et al., 2017).

An apparent solution to issue (iv) would be to increase the elevation of the model top. However, the sensitivity study in Sect.

4.3 showed, that this does not lead to a decrease in the MSE of ICAR or ICAR$_{\mathrm{CP}}$ time series at the alpine weather stations. The reason for the deteriorating performance with increasing model top height is potentially found in issue (v), the influence of divergences in the forcing wind field on the ICAR wind field. Future research could focus on developing a method to prepare the forcing wind field in a way that minimizes the negative effects on the ICAR wind field.

While the relative variability of average daily precipitation amounts related to synoptic weather patterns, $\overline{P_{24h}}(\mathrm{wp}, \mathrm{ws})$, are

reproduced similarly well both by ICAR$_{\mathrm{CP}}$ and ERAI (see Fig. 11a), absolute amounts of $\overline{P_{24h}}(\mathrm{wp}, \mathrm{ws})$ are largely underestimated by ERAI (up to on average $17\,\mathrm{mm}$). This underestimation is far less pronounced in ICAR, resulting in a median SS$_{\mathrm{MSE}}$ of $0.74$ when modeling $\overline{P_{24h}}(\mathrm{wp}, \mathrm{ws})$ (see Fig. 11b)

## 6   Conclusions

In this study, simulations with ICAR were found to provide added value over ERAI for $24\mathrm{h}$ accumulated precipitation on the

South Island of New Zealand for alpine weather stations. In contrast to the almost consistently positive results found for the alpine weather stations, ICAR provides no added value over ERAI for coastal weather stations. A comparison of average and seasonal precipitation patterns of an operational gridded rainfall data set with ICAR showed good agreement. Grouping the available data according to Froude number revealed that stable atmospheric conditions with higher degree of flow linearity lead to higher skill scores, and that ICAR provides added value over ERAI even for days with near-stable conditions and stable days

with lower flow linearity. A grouping according to the synoptic situation showed that values of SS$_{\mathrm{MSE}}$ are generally high for weather patterns associated with flow approximately perpendicular to the alpine range, such as the T, TNW and W patterns, and lowest for weather patterns exhibiting flow parallel to the Southern Alps (NE and HW). While ICAR in principle does not require observations for tuning, the model top for this study was determined with a sensitivity analysis. All other settings could be adopted from default. With the adjusted model top, however, consistent added value for stations in complex topography was

found, with a reduction of the median error by $30\,\%$. Clear improvement may be expected on further site-specific tuning to observations as routinely performed in regional climate model based downscaling. Further research on how ICAR fields are influenced by the forcing data set will be necessary.





*Data availability.* The data sets, ICAR configuration, DEM and forcing files the presented results are based on are available as download from a zenodo repository (Horak et al., 2018). Due to licensing restrictions VCSR data is not included in the repository.

*Competing interests.* The authors declare no competing interests.

5  *Acknowledgements.* The research presented has been funded by the Austrian Science Fund (FWF) grant 28006-N32. The computational results have been achieved with the high-performance computing support from Cheyenne (doi:10.5065/D6RX99HX) provided by NCAR's Computational and Information Systems Laboratory, sponsored by the National Science Foundation and with the HPC infrastructure LEO of the University of Innsbruck. The National Center for Atmospheric Research is sponsored by the US National Science Foundation. Furthermore the authors thank the National Institute of Water and Atmospheric Research, New Zealand, and in particular Christian Zammit, for

10  providing support, the weather pattern classifications, the VCS gridded rainfall data set and the data from the weather stations as specified in Table 1. Research on Brewster Glacier is supported by the Department of Geography, University of Otago, New Zealand; the National Institute of Water and Atmospheric Research (Climate Present and Past CLC01202); and the Department of Conservation under concession OT-32299-OTH. The following open-source libraries were employed to perform the data processing and analysis presented in this study: `numpy` (van der Walt et al., 2011), `pandas` (McKinney et al., 2010), `xarray` (Hoyer and Hamman, 2017), `matplotlib` (Hunter, 2007) and `cartopy` (Met Office, 2010).



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
