# Peer review of "Assessing the Added Value of the Intermediate Complexity Atmospheric Research Model (ICAR) for Precipitation in Complex Topography"

_Hydrology and Earth System Sciences, 2018_

## Referee Comment (RC1) · Leutwyler (Referee) · 17 Dec 2018

In their manuscript Horak et al. assess the skill of the Intermediate Complexity Atmospheric Research Model (ICAR) for downscaling mean precipitation amount, in a domain located over the South Island of New Zealand. Model evaluation is performed using established techniques, a range of observational datasets and two skill scores. Their main findings are: (a) ICAR provides additional skill over the main Alpine ridge, while results over coastal stations are deteriorated. (b) Added value is typically largest for stable upstream flow, impinging on the ridge at a 90° angle. These results seem related to the model's roots, which is built on linear theory of orographic precipitation.

[Figure]

The article is generally well written and suited for publication in HESS (also for GMD). I particularly appreciated its modest and plain language. All review criteria are met, and I did not detect major scientific flaws, considering the manuscripts scope.

Minor Comments

(1) P2L8: "While dynamic downscaling results in a self-consistent set of atmospheric fields, the computational cost required for the fine spatial and temporal grid spacing is high, especially for long-term simulations or sensitivity studies."

This sentence would benefit from perspective. For example, for a similar computational domain we would achieve about 240 simulation days per day when running COSMO on a single node, equipped with a P100 GPU (Leutwyler et al., 2016; Fuhrer et al., 2018, I am not implying that you should cite my studies, but used then because I know the numbers by heart). How does ICAR compare these benchmarks?

(2) P2L12: "to a lesser extent, to dynamic downscaling as well" I don't fully understand the statement in this fragment. Please elaborate on the stationary assumptions in dynamical downscaling, and how precisely this is overcome in ICAR.

(3) Section 2.1: Adding a few plain language sentences how ICAR works and how the approach differs from dynamical downscaling would aid the wider audience. Additionally, a concise summary about linear theory of orographic precipitation and how it is incorporated into ICAR would help. I had to read Gutman et al. (2016) to understand this Section.

(4) Section 4.1: Maybe it would be good to discuss the known biases for mean precipitation in ERAI and outline weather it is difficult to beat it.

(5) Section 4.3 (a) Unfortunately, the chosen calibration period overlaps with the analysis period and employs the same stations. Cross-validation with other periods or station replacement would make the arguments more robust. (b) "Potential reasons for the observed behavior are discussed in Sect. 5." → That statement is a bit mislead-
ing, since in Section 5 you only say that the question remains open. (c) I am skeptic if the results at 2.5 km and 4 km are substantially different from each other. (d) A devil's advocate could argue that ICARCP mainly improves skill over ICAR because the latter underestimate precipitation amount (see P30L27). I.e., could the same skill be achieved by adding random noise with the right magnitude? Does ICAR beat ERAI too? Please elaborate (here or in Section 2.6) to justify your choice to add interpolated conv. precip. from ERAI.

(6) Section 4.7 Why is the underlying dataset changed to NCEP/NCAR?

(7) Section 5 (a) 1st paragraph: It might be worthwhile to elaborate on how these results relate to linear theory of orographic precipitation. (b) P30L6: "Therefore these two instances are considered as outliers." I think there is a problem here.

Suggestions for optional extensions

(1) Downscaling low-resolution global climate simulations (rather than re-analysis), along major mountain ridges could more evidently illustrate the added value of the approach.

(2) From an application/user point of view, employing the outlined techniques to obtain higher-resolution fields is still a somewhat cumbersome procedure. It will therefore only be performed operationally if the added value is rather substantial. Therefore it would be interesting to see the added value over low-resolution precipitation climatologies such as, e.g., GPCC or GPCP..

Technical comments

P1L1: climate downscaling → downscaling techniques P1L7: the eleven-year period from 2007 to 2017 → an eleven-year period, ranging from 2007 until 2017 P1L9: diagnosed → assessed P1L14: In the abstract, I would use a more general term for "flow of higher linearity" P1L17: tuning → calibration (tuning has a negative connotation). Same applies to the rest of the manuscript. P2L21: Maybe add weather generators to

[Figure]

the discussion? P2L31: due → emerging from P3L23: storing → stores P4L5: no data are → no observations are P5L7: ERAI have → ERAI employs (I think ERA-Interim reanalysis is singular). Also P4L11 P5L19: "convective precipitation from the ERAI" Add the name of the field. Also, add a reference to your Table 1. P6L5: New Zealand, P6L7: ranges → maybe "ridges"? P7L12: In case of → For P9L10: Move sentence " The aim is not a downscaling ..." to end of paragraph P9L27: HSS is defined as → The HSS P12L4: I relate "occurrence" to precipitation frequency. Maybe better use magnitude? P15L5: For lazy or tiered readers it might be helpful to re-state that VCSR are the observations. P16L6ff: Maybe indicate which months these seasons are (DJF ..)? P19L15: Cloud you add these regions to Fig. 1? P29L9-17: (a) Maybe move this Paragraph to Section 3.2? (b) Does undercatch not affect HSS(P>50)? P29L9-33: I would move the caveats to another place such that the paragraph currently starting at L34 follows after the current L8. P30L21-24: Could you elaborate why you think this issue is a likely candidate?

Table 1: Outline in caption where the uncertainty estimates come from (+/- 0.1). Figure 2: Are these MSE of the annual sums (Add to the caption)? Maybe add the mean values so the results can be put into perspective. Table 2: These are mm/day (e.g. RMSE (mm)), correct? Figure 5: (a) NIWA (top-left) → VCSR (b) Maybe mean magnitude over land to panels? Figure 6: Why do the no. samples (circles) differ among the various thresholds in HSS? Explain in the caption.

Best regards David Leutwyler

References

Fuhrer, O., T. Chadha, T. Hoefler, G. Kwasniewski, X. Lapillonne, D. Leutwyler, D. Lüthi, C. Osuna, C. Schär, T. C. Schulthess, and H. Vogt (2018), Near-global climate simulation at 1 km resolution: establishing a performance baseline on 4888 gpus with cosmo 5.0, Geosci. Model Dev., 11 (4), 1665–1681, doi:10.5194/gmd-11-1665-2018.

Leutwyler, D., O. Fuhrer, X. Lapillonne, D. L ÌLuthi, and C. Schär (2016), Towards

european-scale convection-resolving climate simulations with gpus: a study with cosmo 4.19, Geosci. Model Dev., 9 (9), 3393–3412, doi:10.5194/gmd-9-3393-2016.

---

## Referee Comment (RC2) · Anonymous Referee #2 · 5 Feb 2019

This manuscript assesses the added value of ICAR relative to coarse reanalysis for estimating precipitation in complex topography. Not yet widely evaluated, ICAR is a promising tool for a range of applications. The methods in this study are robust and the conclusions are important. The manuscript should be a valuable contribution to the literature.

Comments

1. In terms of the manuscript structure, it seems a bit unusual to have a combined "Methods and Results" section (4). I can see why the manuscript was structured as it is, but I wonder if it could be rationalised at all. Could there be benefits from a

more "traditional" separation of methods and results? For example, the major sections could go something like: 1. Introduction 2. Study Area and Data 3. Methods 3.1 ICAR Overview and Setup 3.2 Evaluation Strategy 3.3 Skill Scores and Significance Tests 3.4 Flow Linearity (explaining how flow linearity and stability are calculated) 3.5 Weather Patterns (explaining dataset with figure of weather patterns) 4. Results (as currently structured but removing the methods now described in the previous section) 5. Discussion 6. Conclusions

This is just a possibility; there could be a better way. I would also suggest checking the manuscript for repetition and trying to minimise the amount of "referencing forward" (i.e. sometimes it is not necessary to say "X will be discussed in Section Y").

2. One specific point on structure is that the optimal model top height is stated in Section 2.3, before the results from the sensitivity test are presented. This should be avoided I think. It is already stated in the relevant part of the results section (i.e. on model top sensitivity) that the remainder of the evaluation uses the optimal model top.

3. Also regarding the model top sensitivity, it would be interesting to contextualise the variation in ICAR performance shown in Figure 2 by providing the equivalent MSE for ERAI. This could be as a horizontal line on Figure 2 or just stated in the text. I.e., even for the model tops leading to larger errors, do they still outperform ERAI overall?

4. It is mentioned in the discussion (P29 L19-20) that higher model tops lead to lower precipitation. Does this apply across the full range of model tops tested? Would there be any value in adding a second panel to Figure 2 showing mean bias for the different model tops? I.e. given that ICAR is generally low-biased for the Alpine stations, does a 2.5 km model top lead to reduced bias (even if the MSE is little different from 4 km)? Or does ICAR become high-biased with a 2.5 km top?

5. In several places in the manuscript, results are discussed but the corresponding figures/tables are "not shown". This includes the relationship between model top and precipitation magnitude mentioned above, as well as seasonal averages at Alpine sta-

tion locations (P16 L13-18). The latter case I found confusing initially, as the ERAI seasonality is reasonable-ish in Figure 5 but criticised in the text based on station locations (P16 L17-18). I would suggest considering whether some of the "not shown" figures/tables should be put into a supplement or whether references to them are necessary.

6. It may be useful to provide the season definitions used in Section 4.5 (i.e. which months).

7. The panels in Figure 5 are quite small so it is difficult to make out much of the detail. The overall improvement of ICAR over ERAI is clear though. It would be interesting to see a version of the figure zoomed in on the Alpine range, but perhaps this could be in future work.

8. Figure 3b has a spelling error - "coastal".

9. The boxplots for near-stable conditions in Figure 7c and 7d are quite different. What could be the reason for this?

10. There are a few places where the wording and grammar could be a little bit tighter. For example, sometimes "trend" is used when something like "pattern" might be better. There are other examples too, such as the first three sentences of the paragraph beginning on L15 on P20. The manuscript is generally fairly well written, but I would suggest that the authors check the wording and grammar throughout when making revisions.

11. In the abstract and discussion it is mentioned that ICAR can reduce MSE by up to 53%. If this is the maximum reduction, what is the mean/median? This may be worth including to give the "overall" picture.

12. It could be mentioned again in the discussion/conclusion that a comparison of ICAR and WRF (or a similar model) might also be interesting for this study area. This might help us to understand some of the possible factors limiting ICAR performance discussed in Section 5. It would also give an idea of the relative performance gain from

using WRF (if any) in a different climatic context to that tested in Gutmann et al. (2016).

---

## Referee Comment (RC3) · Anonymous Referee #3 · 15 Feb 2019

Overview This well-written manuscript details a comparison between ERA-Interim and ICAR at generating precipitation over New Zealand's south island. They find that ICAR adds value over ERA Interim at most alpine locations, but not at coastal stations. They additionally tease apart ICAR performance during different flow regimes (identified by the Froude number) and during different weather regimes (identified through synoptic patterns). The work is useful and complete, and I have only minor comments, enumerated below.

Specific Comments P. 3, l. 27-29: During my first read through of the manuscript this sentence made me question how this replacement of unstable locations/times with

[Figure]

weakly stable locations/times impacts ICAR's performance (since it's very unphysical). Some comment here or perhaps in the introduction about application of ICAR during unstable conditions (referring to section 2.6, which is how it is handled in this manuscript), and how/where this factor limits ICAR's use, is warranted.

P. 5, l. 10: '6 h h' the second h is a mistake

P. 5, l. 21-24: I found the way this is notated to be somewhat confusing. I think the reason the authors are using the nomenclature 'ICARcp' to replace P(t) (i.e., ICAR precipitation added to ERA Interim convective precipitation regridded through bilinear interpolation to the 4km grid) is because it's basically ICAR plus convective precipitation. But this seems more complicated than necessary – why not use P(t) and Pi(t) throughout the text? If the authors insist on keeping ICARcp and ICAR then they should use this nomenclature in equation 1 and include a sentence explaining the nomenclature after the equation.

P. 7 l. 12: 'In case of the coastal weather stations,...' is awkward.

P. 8, caption of Table 1, last sentence: 'north respectively south' should read 'north and south, respectively'

P. 12, l. 4: 'performs very similar' should read 'performs very similarly'

P. 13: Fig 3 panel b: coastal is misspelled in title.

P. 14, table 2 caption, last sentence, asterisk is misspelled.

P. 16, l. 16-17: It's unclear to me exactly what this sentence is describing since the figure is not shown; does this mean that the amplitude of the seasonal cycle is too small in ICAR or more generally that ICAR underestimates climatological precipitation at some locations? More discussion is warranted and perhaps this figure should be included in the manuscript.

P. 16, l 23-24: Is there any reason to think that the correspondence in seasonal errors

between ICAR and ERA-I (i.e., that both have largest errors in summer and smallest in winter) is causal? That is, since ERA-Interim provides lateral boundary conditions for ICAR? P. 17, Figure 5: masking the ICAR and ERAI values over the ocean would be less distracting (since there is no 'truth' over the ocean, anyways).

P. 19, l. 11: What percentage of the crest of the southern Alps is over 1500m? Based on Fig. 1 it seems closer to 1000m would be a somewhat more appropriate height to use in the calculation of Froude number; are the results pertaining to the Fr<1 and Fr>1 cases sensitive to this mountain height?

P. 25, second sentence in Fig 9 caption: This sentence is poorly worded.

P. 26, l. 12-13: This sentence is poorly worded.

P. 29. L18-23: Can the authors speculate why there is this sensitivity to model top height?

P. 29, L. 21: 'estimation the model' is missing 'of'

P. 30, L. 9-13: This paragraph should be expanded for clarity (i.e., rather than saying 'solution to issue (iv) it would be helpful to repeat the description of the issues).

---

## Referee Comment (RC4) · Trevor Carey-Smith (Referee) · 21 Feb 2019

I have not been able to complete a proper review of this manuscript due to other more pressing time commitments. In lieu of a full review, I have a few minor comments that can be easily addressed by the authors:

1. The VCS gridded data set is a thin-plate spline based gridded observation set using a mean rain surface as a covariate. This surface is derived using "expert judgement" from observations and elevation, not from "physics-based regional climate modelling" as stated on line 3 page 7.

[Figure]

2. For the NCD database, a link https://cliflo.niwa.co.nz could be included.

3. The caption for Figure 4 has the elevation of Albert Burn as 120m it should be 1280m.

From my cursory read of this manuscript, I think it will be of interest to readers and is well worth publishing.

---

## Author Comment (AC1) · 29 Mar 2019

**Response to Reviewer 1**

**Abbreviations:**

> AR    Author Response (Johannes Horak)
>
> RC    Reviewer Comment

**RC:** In their manuscript Horak et al. assess the skill of the Intermediate Complexity Atmospheric Research Model (ICAR) for downscaling mean precipitation amount, in a domain located over the South Island of New Zealand. Model evaluation is performed using established techniques, a range of observational datasets and two skill scores. Their main findings are: (a) ICAR provides additional skill over the main Alpine ridge, while results over coastal stations are deteriorated. (b) Added value is typically largest for stable upstream flow, impinging on the ridge at a 90∘ angle. These results seem related to the model's roots, which is built on linear theory of orographic precipitation. C1

The article is generally well written and suited for publication in HESS (also for GMD). I particularly appreciated its modest and plain language. All review criteria are met, and I did not detect major scientific flaws, considering the manuscripts scope.

> **AR:**
>
> We thank the reviewer for his effort, and are very appreciative of the detailed comments and criticism of our manuscript. We took every comment very seriously and adjusted the manuscript accordingly.
>
> Our efforts to address one comment regarding the flow-linearity analysis led to the discovery of an error in the underlying data set. We redid the entire analysis with the correct data and updated the affected parts in the methods section and in the discussion. However, the essential characteristics of the results have not changed.
>
> Please find a detailed response to every comment below.
>
> **Corrections to the manuscript independent of the RCs:**
>
> **P5L8:** We found that the list of fields contained in the forcing file was incomplete. We added the two missing fields, the sentence now reads:
>
> "The assembled ICAR forcing file contains ERAI zonal and meridional winds U and V, potential temperature $\Theta$, pressure p, specific humidity $q_v$ , **cloud liquid water mixing ratio $q_c$ , cloud ice water mixing ratio $q_i$** and surface pressure $p_0$ at each 6 h forcing time step and every grid point within the domain."
>
> **P32L14:** The list of employed open-source libraries was incomplete. We added the missing library. The sentence now reads:
>
> "numpy (van der Walt et al., 2011), pandas (McKinney et al., 2010), xarray (Hoyer and Hamman, 2017), matplotlib (Hunter, 2007), cartopy (Met Office, 2010) **and salem (Maussion et al., 2019**)."

**Minor Comments**

**RC:** (1) P2L8: "While dynamic downscaling results in a self-consistent set of atmospheric fields, the computational cost required for the fine spatial and temporal grid spacing is high, especially for long-term simulations or sensitivity studies." This sentence would benefit from perspective. For example, for a similar computational domain we would achieve about 240 simulation days per day when running COSMO on a single node, equipped with a P100 GPU (Leutwyler et al., 2016; Fuhrer et al., 2018, I am not implying that you should cite my studies, but used then because I know the numbers by heart). How does ICAR compare these benchmarks?

**AR:**

The South Island of New Zealand ICAR simulations with 12 vertical levels, for instance, when run on one node of NCAR's Cheyenne cluster (with 36 2.3-GHz Intel Xeon E5-2697V4 Broadwell cores) have a ratio of about 10.5 simulated years per day (on average 200 core hours per simulated year). The following barplot shows the average number of core hours required by ICAR to simulate one year for the South Island of New Zealand domain in dependence of the number of vertical levels.

[Figure]

While we did not run WRF simulations for our study, Gutmann et al. 2016 did so. They found, that, depending on (but not only) the number of vertical levels and chosen microphysics parametrisation, ICAR speeds up simulations by a factor of 140. E.g. one simulated year for the Colorado domain as specified in Gutmann et al. (2016) and a WRF setup as given in Rasmussen (2014) required ~40,000 core hours (if the simulation were run on one CPU core only). ICAR, on the other hand, completes the simulation after ~300 core hours.

To clarify and lend perspective we added a sentence to P2L22-26 that references the findings of the Gutmann 2016 paper in the context of ICARs computational frugality. We also replaced the erroneously used term "linear theory of orographic precipitation" with the correct term "linear mountain wave theory". Please be aware that the updated paragraph includes changes made due to another comment as well (shown as non-bold, orange text):

"The Intermediate Complexity Atmospheric Research model (ICAR; Gutmann et al., 2016) offers a computationally frugal and physics-based alternative that does not rely on measurements with **linear mountain wave theory** as its theoretical foundation. In comparison to other downscaling

approaches of intermediate complexity (e.g. Sarker, 1966; Rhea, 1977; Smith and Barstad, 2004; Georgakakos et al., 2005), ICAR is a more general atmospheric model that requires fewer simplifying assumptions about the state of the atmosphere, such as spatial and temporal homogeneity of the background flow. Furthermore, in contrast to the linear theory of orography precipitation (LOP; Smith and Barstad, 2004), ICAR considers a detailed vertical structure of the atmosphere and employs a complex microphysics scheme as opposed to the characteristic timescales for cloud water conversion and hydrometeor fallout of the LOP. **With regards to dynamical downscaling, in particular the Weather Research and Forecasting model, Gutmann et al. (2016) have shown that ICAR may reduce the required computational time for one simulated year for a domain in the Western United States by a factor of at least 140.**"

**RC:** (2) P2L12: "to a lesser extent, to dynamic downscaling as well" I don't fully understand the statement in this fragment. Please elaborate on the stationary assumptions in dynamical downscaling, and how precisely this is overcome in ICAR.

**AR:**
If, for instance, a dynamical downscaling model is calibrated with measurements this indicates that not all parameters or variables may be inferred from theory or first principles. It follows that the parameters (or even a specific choice of a parametrization over another) determined by the calibration period may not necessarily apply to other periods with altered conditions equally as well. For global climate models, for instance, Maraun et al. (2017) note that "Often, a realistic behaviour is achieved only by tuning the model."
This applies to ICAR as well if empirical parameters of a physical process (i.e. parameters of the microphysics parametrization) are calibrated with measurements. Therefore, dynamical downscaling and intermediate complexity downscaling are both affected by the stationarity assumption if calibrated with measurements. We removed the part of the sentence to avoid insinuating that ICAR, when calibrated with measurements, somehow overcomes the stationarity assumption. The sentence now reads (removed text crossed out):

"Even more problematic, as soon as observation-based training or tuning is applied, the assumption of stationarity is introduced for statistical downscaling  which may not hold under a changing climate (Maraun, 2013; Gutmann et al., 2012)."

**RC:** (3) Section 2.1: Adding a few plain language sentences how ICAR works and how the approach differs from dynamical downscaling would aid the wider audience. Additionally, a concise summary about linear theory of orographic precipitation and how it is incorporated into ICAR would help. I had to read Gutman et al. (2016) to understand this Section.

**AR:**
We rephrased the first and second paragraph of Section 2.1. (P3L11-16 and P3L17-21) to give a better overview of the basic functionality of ICAR, and its main difference from dynamical downscaling. The first paragraph (formerly at P3L11-16) now reads:

"**ICAR (Gutmann et al., 2016) is a three-dimensional atmospheric model based on linear mountain wave theory. As input ICAR requires a digital elevation model and a forcing dataset with 4-D atmospheric variables generated by, for instance, a coupled atmosphere-ocean general circulation model or an atmospheric reanalysis such as ERA-Interim. The forcing dataset should at least contain the horizontal wind components, pressure, temperature and water-vapor mixing ratio, with the possibility to additionally include**

**hydrometeor fields, incoming long and short-wave radiation or the skin temperature of water bodies. ICAR employs linear mountain wave theory to calculate the wind field from the topography information and the horizontal wind components to avoid a numerical solution of the Navier-Stokes equations of motion, the core of dynamical downscaling models. With this wind field, ICAR advects atmospheric quantities, such as temperature and moisture as supplied by the forcing dataset at the domain boundaries. In its standard setup ICAR applies the Thompson microphysical scheme (Thompson et al., 2008), a double moment scheme in cloud ice and rain and a single moment scheme for the remaining quantities to compute the mixing ratios of water vapor, cloud water, rain, cloud ice, graupel and snow.**"

The second paragraph (formerly at P3L17-21) now reads:

"**The classic approach of linear mountain wave theory predicts the wind field based on the topography and the background state of the atmosphere. (Sawyer, 1962; Smith, 1979). With the background state known, its perturbation due to topography is given by a set of analytical equations (Barstad and Grønås, 2006). However, linear theory does not take into account interactions among waves or waves and turbulence, nor transient and non-linear phenomena such as time-varying wave amplitudes, gravity wave breaking or low-level blocking and flow splitting. A basic discussion of the limitations implicit to these assumptions can be found in Nappo (2012). In ICAR, the atmospheric background state is given by the forcing dataset. This yields a time sequence of steady state wind fields between which ICAR interpolates linearly. A detailed description of the model is given in Gutmann et al. (2016).**"

**RC:** (4) Section 4.1: Maybe it would be good to discuss the known biases for mean precipitation in ERAI and outline weather it is difficult to beat it.

**AR:**
A general statement about the performance of ERAI and how hard it is to beat is difficult to make since it depends, among other things, on the particular region of the world that is investigated and the specific factors that influence the local climate. Skill scores alone, in terms of percentage improvement, cannot fully account for how accurate a model is if nothing more is known about the reference model. For this reason we based our evaluation not on skill scores alone. We investigated the ICAR and ERAI precipitation time series at the weather stations as well and compared them directly to measurements. In our region ERAI simulates occurrence well and reproduces the measured time series but underestimates the precipitation magnitude (see Figure 4). This is in stark contrast to, for instance, the Peruvian Andes, another region we are currently investigating. Here ICAR skill scores are positive as well but precipitation occurrence and magnitude is not reproduced at all at some locations. The reason for the positive scores is that the performance of ERAI at these sites is worse. While ICAR is able to correct a little bit towards the measurements, this does not imply that the generated time series are realistic.

For a definitive assessment whether ERAI is difficult to beat, it is necessary to compare ERAI precipitation time series to those of measured at sites of interest. However, this was not the intended aim of the manuscript presented. Nonetheless we believe that Figure 4 gives a representative overview of the capabilities of ERAI with regards to modelling the 24h accumulated precipitation at the sites investigated within the study domain. While the timing of precipitation events is generally well captured by ERAI this is not the case for the magnitudes of the precipitation events.

**RC:** (5) Section 4.3 (a) Unfortunately, the chosen calibration period overlaps with the analysis period and employs the same stations. Cross-validation with other periods or station replacement would make the arguments more robust.

**AR:**

We agree with this assessment. Unfortunately even though ICAR is computationally more efficient than dynamic downscaling, performing, for instance, leave-p-out cross-validation would require extensive computational resources. However, the results suggest that the calibration period (2014-2015) is representative of the full study period (2007-2017) with regards to the presented calibration method. For the simulations with 12 vertical levels, the mean MSE of ICAR shows only little variation on whether the MSE is calculated for the calibration period, the entire study period or the study period excluding the calibration period.

To address this comment we added an additional paragraph to Section 4.3, an additional Panel to Figure 2 and an additional paragraph to the Discussion.

The new paragraph in Section 4.3 and the adapted Figure 2:

**"The mean MSE over all alpine weather stations is almost constant when calculated either for the reference period (2014-2015), the full study period (2007-2017) or the reduced study period, where the reference period is excluded from the time series (2007-2013 and 2015-2017), see Fig. 4c. This result indicates that the reference period is representative of the full study period."**

[Figure]

**Figure 2.** The average (a) MSE (b) $SS_{MSE}$ of ICAR and $ICAR_{CP}$ time series from simulations for the reference period 2014-2015 at alpine weather stations as a function of the chosen model top (in km above topography). Connecting lines serve as guides to the eye. Panel (c) shows the distribution of skill scores for simulations with a model top set 4.0km above topography at alpine weather stations for the reference period (2014-2015), the full study period (2007-2017) and the reduced study period where the reference period has been removed from the dataset (2007-2013 and 2016-2017). The lower boundary of each box indicates the 25th percentile, the upper boundary the 75th percentile and the dashed horizontal line the mean. Whiskers show the minimum and maximum values of the data set.

The new paragraph that we added to the discussion:

**"In this study, the chosen reference period (2014-2015) overlaps with the study period (2007-2017). While ICAR is computationally more efficient than dynamic downscaling, performing, for instance, leave-p-out cross-validation would require extensive computational resources. However, the results suggest that the reference period is representative of the full study period with regards to the presented calibration method: For simulations with the model top set at 4 km, the mean MSE over all alpine weather stations of ICAR shows only little variation on whether the MSE is calculated for the reference period, the study period or the study period excluding the reference period (see Fig. 2c). Furthermore, the variation between the mean MSEs for simulations with different model top settings (Fig. 2b) is larger than the variation between different evaluation periods (Fig. 2c)."**

**RC:** (b) "Potential reasons for the observed behavior are discussed in Sect. 5." ! That statement is a bit misleading, since in Section 5 you only say that the question remains open.

> **AR:**
> Thank you for bringing this to our attention. The second part of the discussion that is concerned with this statement is located in a different paragraph in the discussion section (see P31L11-13). We rearranged the discussion section and included a only recently discovered potential cause (numerical artifact from model top treatment), the corresponding paragraph now reads:
>
> "The sensitivity studies leading to the choice of the model top at 4 km have shown that the model top elevation greatly influences precipitation amounts and in turn the obtained mean squared errors, see Fig. 2. It is not immediately obvious though why precipitation amounts decrease (not shown) and the MSEs deteriorates for higher model tops. **Potential reasons are influences of divergences in the forcing wind field on the ICAR wind field or numerical artifacts arising from the treatment of the model top in ICAR. However, further research is necessary to develop a better understanding of this issue and its causes. Subsequently future studies could focus on finding** a method that allows the estimation of the model top elevation best suited for a domain without relying on measurements, as well as on **investigating** the influence of the choice of the forcing data type (i.e. global or regional reanalyses, GCMs, weather forecast models) and the spatial grid resolution thereof on ICAR dynamics and skill."

**RC:** (c) I am skeptic if the results at 2.5 km and 4 km are substantially different from each other.

> **AR:**
> One major difference between the two runs is that the simulations with a model top at 2.5 km cut off the atmosphere within layers that transport a significant amount of moisture within the domain. This entails a less faithful representation of the moisture content of the atmosphere and may in part lead to unphysical artifacts in the moisture distribution due to the way the model top is treated by ICAR. However, more research is necessary to quantify and understand this effect and how it affects the distribution of precipitation and moisture throughout the domain. While MSEs at alpine sites are similar but lower for simulations with a model top at 4.0 km, a particularly adverse effect is observed with regards to precipitation occurrence (HSS scores with $P_{24h} > 1$ mm). Here, a distinct score decrease is observed at all except one weather station if the model top is set to 2.5 km or lower.

[Figure]

**RC:** (d) A devil's advocate could argue that ICARCP mainly improves skill over ICAR because the latter underestimate precipitation amount (see P30L27). I.e., could the same skill be achieved by adding random noise with the right magnitude?

> **AR:**
> We tested this hypothesis and found that the addition of random noise to ICAR precipitation time series is not able to achieve the same mean skill as $ICAR_{CP}$. Moreover, even adding random noise only to days where ICAR predicts non-zero precipitation (semi-random noise) does not lead to a higher skill than is achieved by $ICAR_{CP}$.

[Figure]

**RC:** Does ICAR beat ERAI too?

> **AR:**
> We added an additional panel to Figure 2 that indicates the mean $SS_{MSE}$ at alpine weather stations achieved for each model top setting during the calibration period. ICAR is able to outperform ERAI for model tops at 1.5 km, 2.5 km and 4.0 km.
>
> The updated Figure 2, the additional panel referenced above is panel b:

[Figure]

**Figure 2.** The average (a) MSE (b) SS$_{MSE}$ of ICAR and ICAR$_{CP}$ time series from simulations for the reference period 2014-2015 at alpine weather stations as a function of the chosen model top (in km above topography). Connecting lines serve as guides to the eye. Panel (c) shows the distribution of skill scores for simulations with a model top set 4.0km above topography at alpine weather stations for the reference period (2014-2015), the full study period (2007-2017) and the reduced study period where the reference period has been removed from the dataset (2007-2013 and 2016-2017). The lower boundary of each box indicates the 25th percentile, the upper boundary the 75th percentile and the dashed horizontal line the mean. Whiskers show the minimum and maximum values of the data set.

**RC:** Please elaborate (here or in Section 2.6) to justify your choice to add interpolated conv. precip. from ERAI

> **AR:**
> ICAR is not able to model convective precipitation by itself in the setup used. Since ERAI does simulate convective precipitation and store the value in a separate field it seems a reasonable choice to use this additional information provided by the forcing dataset to improve the precipitation fields and time series simulated by ICAR. This is elaborated in Section 2.6, P5L18-20. Furthermore, it is a common technique to use convective or large scale precipitation from the forcing dataset this way, compare, for instance Roth et al. (2018) or other studies where the downscaled precipitation is a composite of precipitation generated by the downscaling model and the forcing for types of precipitation the applied model cannot account for (Jarosch et al. 2012; Weidemann et al. 2013 and Paeth et al. 2017).
>
> To clarify we added these references to the manuscript. Please note that the updated paragraph contains an additional sentence added due to another RC (orange, non-bold text). Section 2.6 P5L24 now reads:
>
> "where in the following the P(t) time series is referred to as ICARCP and $P_I(t)$ as ICAR. **This is a common technique that allows to include types of precipitation not accounted for by the downscaling model (e.g. Jarosch et al., 2012; Weidemann et al., 2013; Paeth et al., 2017; Roth et al., 2018).** Table 1 shows the mean annual precipitation at each site for ICAR$_{CP}$ and ERAI, as well as the ratio of ERAI convective precipitation to ERAI total precipitation."

**RC:** (6) Section 4.7 Why is the underlying dataset changed to NCEP/NCAR?

> **AR:**

The cluster analysis yielding the weather-patterns was not performed by us but by Kidson (2000), who employed the NCEP/NCAR dataset. We rephrased for clarity, it now reads:

"For the underlying cluster analysis, **Kidson (1994a) employed** the NCEP/NCAR 40-year reanalysis dataset (Kalnay et al., 1996) between January 1958 and June 1997 ."

**RC:** (7) Section 5 (a) 1st paragraph: It might be worthwhile to elaborate on how these results relate to linear theory of orographic precipitation.

**AR:**
The linear theory of orographic precipitation (LOP) is, while connected to ICAR via the common basis of linear mountain-wave theory, not directly related to the results presented here. One fundamental difference is that the LOP, unless adapted as in, for instance, Jarosch (2012), is only able to consider a homogeneous background state across the entire domain. Similarly, unless adapted as in Barstad and Schüller (2011), information about the vertical structure of the atmosphere is, compared to ICAR, very basic. Another key difference is the use of a complex microphysics scheme (Thompson, 2008) in ICAR, while the LOP considers characteristic timescales for cloud water conversion and hydrometeor fallout. A comparison between the LOP and ICAR would be of interest, but outside of the scope of our manuscript.

To highlight the differences between the two models we modified P2L22-26 in the introduction. We also replaced the erroneously used term "linear theory of orographic precipitation" with the correct term "linear mountain wave theory". Please be aware that the updated paragraph includes changes made due to another comment as well (orange, non-bold text). The updated paragraph now reads:

"The Intermediate Complexity Atmospheric Research model (ICAR; Gutmann et al., 2016) offers a computationally frugal and physics-based alternative that does not rely on measurements with **linear mountain wave theory** as its theoretical foundation. In comparison to other downscaling approaches of intermediate complexity (e.g. Sarker, 1966; Rhea, 1977; Smith and Barstad, 2004; Georgakakos et al., 2005), ICAR is a more general atmospheric model that requires fewer simplifying assumptions about the state of the atmosphere, such as spatial and temporal homogeneity of the background flow. **Furthermore, in contrast to the linear theory of orography precipitation (LOP; Smith and Barstad, 2004), ICAR considers a detailed vertical structure of the atmosphere and employs a complex microphysics scheme as opposed to the characteristic timescales for cloud water conversion and hydrometeor fallout of the LOP.** With regards to dynamical downscaling, in particular the Weather Research and Forecasting model, Gutmann et al. (2016) have shown that ICAR may reduce the required computational time for one simulated year for a domain in the Western United States by a factor of at least 140."

**RC:** (b) P30L6: "Therefore these two instances are considered as outliers." I think there is a problem here

**AR:**
Following up other suggestions of the reviewer led us to discover that some ERAI grid points used for the flow linearity analysis were at the wrong locations (too close to the coast). We corrected this and redid the entire analysis. With the updated plots the added value of ICAR over ERAI for higher flow linearity and atmospheric stability is now more evident and the corresponding outliers have vanished. For more details see comment "P19L15: Cloud you add these regions to Fig. 1?" farther below.

**Suggestions for optional extensions**

**RC:** (1) Downscaling low-resolution global climate simulations (rather than re-analysis), along major mountain ridges could more evidently illustrate the added value of the approach.

> **AR:**
> We agree that this would indeed be a worthwhile analysis, it is outside of the scope of the presented manuscript. Additionally, some of the presented methods appear to be difficult to apply to global climate simulations, in particular the weather pattern analysis and the dependency of model performance on flow linearity.

**RC:** (2) From an application/user point of view, employing the outlined techniques to obtain higher-resolution fields is still a somewhat cumbersome procedure. It will therefore only be performed operationally if the added value is rather substantial. Therefore it would be interesting to see the added value over low-resolution precipitation climatologies such as, e.g., GPCC or GPCP..

> **AR:**
> We agree that this is a potentially fruitful avenue for further investigations. However, generally dynamic and statistical downscaling methods alike are generally tested for whether they actually improve over the employed forcing dataset (e.g. Jarosch 2012, or, for a review, Torma et al. 2015). ICAR is a relatively new model and, as mentioned in the introduction, this has not been established yet at the weather station level.

**Technical Comments**

Technical comments

**RC:** P1L1: climate downscaling => downscaling techniques

> **AR:** rephrased as suggested.

**RC:** P1L7: the eleven-year period from 2007 to 2017 => an eleven-year period, ranging from 2007 until 2017

> **AR:** rephrased as suggested.

**RC:** P1L9: diagnosed=> assessed

> **AR:** rephrased as suggested.

**RC:** P1L14: In the abstract, I would use a more general term for "flow of higher linearity"

> **AR:** Exchanged "flow of higher linearity" for "flow linearity".

**RC:** P1L17: tuning => calibration (tuning has a negative connotation). Same applies to the rest of the manuscript.

> **AR:** Exchanged tuning for fitting variations of calibration throughout the manuscript.

**RC:** P2L21: Maybe add weather generators to the discussion?

> **AR:** While weather generators are functionally different from regression models, they do fall in the statistical downscaling category.

**RC:** P2L31: due => emerging from

    **AR:** Rephrased accordingly

**RC:** P3L23: storing => stores

    **AR:** Corrected accordingly

**RC:** P4L5: no data are => no observations are

    **AR:** Rephrased accordingly

**RC:** P5L7: ERAI have => ERAI employs (I think ERA-Interim reanalysis is singular).

    **AR:** Corrected accordingly

**RC:** P4L11 P5L19: "convective precipitation from the ERAI" Add the name of the field. Also, add a reference to your Table 1.

    **AR:** Name and ID of the ERAI field was added and we referenced Table 1 at the end of the paragraph. Please note that the updated text as shown below includes an additional sentence due to another RC (orange, non-bold text).

    Section 2.6 now reads:

    The ICAR configuration for this study, as described in Sect. 2.2, is able to model orographic precipitation and, at least in part, precipitation driven by the synoptic scale. To account for convective precipitation, convective precipitation from ERAI **(field name: cp, parameter ID: 143)**, $P_{CP}$, is resampled to the ICAR timestep and bilinearly interpolated in space to the sites of interest and then added to the ICAR precipitation time series $P_I$:

$$P(t) = P_I(t) + P_{CP}(t), \quad\quad\quad\quad\quad (1)$$

    where in the following the $P(t)$ time series is referred to as $ICAR_{CP}$ and $P_I(t)$ as ICAR. This is a common technique that allows to include types of precipitation not accounted for by the downscaling model (e.g. Jarosch et al., 2012; Weidemann et al., 2013; Paeth et al., 2017; Roth et al., 2018). **Table 1 shows the mean annual precipitation at each site for $ICAR_{CP}$ and ERAI, as well as the ratio of ERAI convective precipitation to ERAI total precipitation.**

**RC:** P6L5: New Zealand

    **AR:** We rephrased the first sentence.

**RC:** P6L7: ranges => maybe "ridges"?

    **AR:** We rephrased the paragraph, it now reads:

    "This study focuses on the Southern Alps **of New Zealand** located in the southwestern Pacific Ocean. The **Southern Alps are** oriented southwest-northeast and run almost parallel to the western coast of the South Island. **They are** approximately 800 km long and 60 km wide, **extend** across a latitude range from 41° S to 46° S and consist of a series of ranges and basins (Barrell et al., 2011)."

**RC:** P7L12: In case of => For

**AR:** Rephrased as suggested.

**RC:** P9L10: Move sentence " The aim is not a downscaling ..." to end of paragraph

**AR:** Moved to the end of the paragraph.

**RC:** P9L27: HSS is defined as The HSS

**AR:** Rephrased as suggested

**RC:** P12L4: I relate "occurrence" to precipitation frequency. Maybe better use magnitude?

**AR:** The HSS for thresholds of P24h > 1mm may be seen as an indicator of whether ICAR$_{CP}$ is better able to model the frequency/occurrence of wet or dry days in comparison to ERAI. Higher tresholds, on the other hand, are more indicative of whether ICAR$_{CP}$ improves the frequency of larger precipitation events over ERAI. We exchanged occurrence for frequency for better clarity.

**RC:** P15L5: For lazy or tiered readers it might be helpful to re-state that VCSR are the observations.

**AR:** Rephrased to "The **observation and expert-judgment based** VCSR, ICAR, ICAR$_{CP}$ and ERAI"

**RC:** P16L6ff: Maybe indicate which months these seasons are (DJF..)?

**AR:** We added abbreviations of the months that are associated with each season to the second paragraph of Section 4.5 and the caption of Figure 5:

"The seasonal variations of precipitation as derived from the VCSR data set (Fig. 5b-e) are best reproduced by ICARCP (Fig. 5l-o). However, the improvements over the corresponding ICAR patterns 5g-j) are small and the remainder of this paragraph applies to ICAR and ICARCP alike. When comparing VCSR and ICARCP the similarities are largest for winter (**JJA**, Fig. 5h and 5m) and summer (**DJF**, Fig. 5e and 5o). The differences increase for the remaining seasons, with the Southern Alps being particularly affected. For autumn (**MAM**), VCSR shows the precipitation as below average (Fig. 5b) while ICARCP indicates above average precipitation (Fig. 5l). For spring (**SON**), on the other hand, VCSR shows an increase in precipitation throughout the Southern Alps (Fig. 5d) but ICARCP shows the central part of the Southern Alps as drier than on average (Fig. 5n)."

Figure 5. The top four panels show patterns of P24h averaged over 2007–2016 for VCSR (left), ICAR (second column), ICARCP (third column) and ERAI (right) over the South Island of New Zealand and surrounding ocean. Rows two to five show seasonal deviations of the all-year average patterns, for autumn (**MAM**, second row), winter (**JJA**, third row), spring (**SON**, fourth row) and summer (**DJF**, bottom). Each panel shows the coastline and the 1000 m MSL contour line of the topography.

**RC:** P19L15: Cloud you add these regions to Fig. 1?

**AR:** We want to explicitly thank the reviewer for this comment as it revealed that some ERAI gridpoints used to determine the flow linearity were not within the test region (they were closer to the coast) and that the length of the test regions was erroneously stated as 1000 km when it should be 500 km. With the new test regions, furthermore, the maximum value of $\kappa$ where still enough data points remained in the near stable category to calculate SS$_{MSE}$ is $375 \cdot 10^{-5}$ s$^{-1}$.

However, the characteristics of the results remain essentially the same, with only minor effects on their discussion and presentation.

**This entails the following changes in Section 4.6 and in the discussion:**

**P19L13:** testregion dimensions corrected:
"... and is about 200 km wide, **500** km long and 1500 m high"

**P19L6-7:** the upper limit of $\kappa$ has changed:
"…the value of $\kappa$ is varied between $25 \cdot 10^{-5}$ s$^{-1}$ and **375** $\cdot 10^{-5}$ s$^{-1}$ in steps of $25 \cdot 10^{-5}$ s$^{-1}$."

**P20L15:** the number of days that fulfill the defined criteria has changed and we adjusted the text according to a criticism of reviewer 2:
"Of the 4018 days in the eleven-year study period, **1847 fulfill the criteria stated above**. A detailed overview **of** the distribution of these days among the three categories in dependence of $\kappa$ is given in Table 3."

**P20L17-23:** we updated the description of the results:
"The results from Table 3 summarized in Fig. 7 show, that stable atmospheric conditions and Froude numbers larger or equal to unity lead to an increase in median scores for sites in complex topography. This behavior is observed for SS$_{\text{MSE}}$ where the score median increases from **0.33** to **0.58** and, for P$_{24h} > 25$ mm and P$_{24h} > 50$ mm in case of HSS. For P$_{24h} > 1$ mm the **maximum median score is found for stable conditions and F < 1, with the F ≥ 1 regime even yielding a negative median score.**"

**P21:** Table 3 was updated and filled with the correct values.

**P22:** Figure 7 was updated and now shows the correct values.

[Figure]

**Figure 7.** Dependence of SS$_{\text{MSE}}$ and HSS at alpine stations on atmospheric stability and the Froude number regime, calculated for all available data for each value of $\kappa$ (see Table 3). SS$_{\text{MSE}}$ is shown in (a) and HSS for thresholds (b) $P_{24h} > 1$mm, (c) $P_{24h} > 25$mm and (d) $P_{24h} > 50$mm. The x-axis indicates atmospheric stability and Froude number regime. The lower boundary of each boxplot indicates the 25th percentile, its upper boundary the 75th percentile and a black line the median. Whiskers show the minimum and maximum values of the data set.

**P29-30L34-10**: We updated the discussion and included additional Figures to reduce the amount of times secondary results are not shown (as per request of Reviewer 2):

ICAR was found to perform better for upstream flows with Froude numbers larger than unity. This result is not unexpected, since linear theory is the theoretical foundation for ICAR. Therefore, flows of higher linearity lead to increased SSMSE and HSS for thresholds of 25 mm and 50 mm. These results hold even if the method for classifying near-stable or stable days is changed. For instance, using $N^2 \leq 0$ as classification criterion for near-stable days and $N^2 > 0$ for stable days leads to similar results (**see Fig. A2**). **For $SS_{MSE}$ (see Fig. 7a) the spread of scores derived from varying $\kappa$ for near-stable days is large enough to include the median score of the stable days with $F < 1$. Nonetheless, this is only true for $\kappa = 200 \cdot 10^{-5}$ s, in all other cases stable days with $F < 1$ always score higher than near stable days. Stable days with $F \geq 1$, in comparison, always achieve a higher score than the other two categories.** A potential issue with the methodology is the small number of cases in the stable regime with $F \geq 1$ compared to the two other classes (see Table 3). However, $P_{24h}$ on stable days with $F \geq 1$ is three to seven times as high as $P_{24h}$ during the other two classes (**see Fig. A3**). Therefore, while comparably small in number, stable days with $F \geq 1$ contribute above-average amounts of precipitation to the climatology, highlighting the importance of the improvement in skill for this category.

**P6**: Figure 1 with the test regions included now is:

[Figure]

**Figure 1.** The South Island of New Zealand. Shown are the coast (black line), the topography above an elevation of 1000 mMSL (gray shading), glacierized areas (blue shading), the approximate location of the main alpine crest (red line) and the location of test regions (dashed outlines) northwest and southeast of the mainland used to determine flow linearity. The alpine weather stations considered in the evaluation of this study are indicated by white triangles, while coastal weather stations are represented by orange disks. The numbers next to the markers are ordered from lowest to highest weather station elevation and may be used to look up additional information for each station in Table 1.

**RC:** P29L9-17: (a) Maybe move this Paragraph to Section 3.2?

**AR:** While we agree that Section 3.2. would be a fitting place for paragraph P29L9-L17 as well, in this manuscript the uncertainties associated with precipitation measurements are only brought up in the Discussion section. To void unnecessary zig-zag and keep the logical flow of the discussion intact, as proposed by Mensh (2017), we decided to leave the paragraph at its current location.

**RC:** (b) Does undercatch not affect HSS(P>50)?

**AR:** Undercatch does affect HSS(P>50) as well and, as detailed in paragraph P29L9-17, is expected to affect both, the performance of $ICAR_{CP}$ and ERAI.

**RC:** P29L9-33: I would move the caveats to another place such that the paragraph currently starting at L34 follows after the current L8.

**AR:** We agree that the discussion would benefit from a more rigid structure. We therefore moved the general discussion of results up so that it now begins after L8. The caveats are now discussed subsequent to the general discussion of results.

**RC:** P30L21-24: Could you elaborate why you think this issue is a likely candidate?

**AR:** We expanded the corresponding paragraph to elaborate further.

It now reads: "**A potential cause for the observed negative correlation is, that the reflection of mountain waves at the interfaces between atmospheric layers can impact the distribution of orographic precipitation (Barstad and Schüller, 2011). Siler and Durran (2015) found, for instance, that wave reflection at the tropopause may either strengthen or weaken low-level windward ascent, which in turn affects the amount and distribution of orographic precipitation. The outcome was found to depend on the ratio of the tropopause height to the vertical wavelength of the mountain waves. Since ICAR currently does not account for wave reflection, its implementation could therefore lead to improvements in this regard.**"

**RC:** Table 1: Outline in caption where the uncertainty estimates come from (+/- 0.1).

**AR:** We added a short outline to the caption:

"List of weather stations used in this study sorted by their elevation. The table lists station number, elevation z, latitude (lat), longitude (lon), name, average distance downwind of the main crest of the Southern Alps (Δ) based on westerly and northwesterly flow, mean annual precipitation $\overline{P}$ **with the standard deviation both calculated for the years where data was available at the respective weather station**, fraction of convective precipitation in ERAI annual sum $f_{cp}$, length of the time series (l) and number of days removed due to missing entries or failed quality checks ($d_m$). The superscript following the station name indicates the data provider:  NCD (1), NIWA (2) and University of Otago (3). Precipitation data for Larkins and Potts were lineary extrapolated to a full year. Δ was not considered for coastal weathers stations and no values were assigned for Mahanga and Larkins since they lie north and south, respectively, of the main alpine crest."

**RC:** Figure 2: Are these MSE of the annual sums (Add to the caption)? Maybe add the mean values so the results can be put into perspective.

**AR:** Figure 2 shows the average over all the MSE of $P_{24h}$ calculated at each alpine weather stations.

**RC:** Table 2: These are mm/day (e.g. RMSE (mm)), correct?

> **AR:** Correct, we adjusted the units in the column headers for clarity, the header now looks like this:

| No | Name | length (yr) | days with $P_{24h}$ above (%) | | | $SS_{MSE}$ (1) | RMSE (mm day$^{-1}$) | | bias (mm day$^{-1}$) | | HSS (1) | | |
|----|------|-------------|------|------|------|------|------|------|------|------|------|------|------|
| | | | 1mm | 25mm | 50mm | | $ICAR_{CP}$ | ERAI | $ICAR_{CP}$ | ERAI | 1mm | 25mm | 50mm |

Figure 5:

**RC:** (a) NIWA (top-left) -> VCSR

> **AR:** We exchanged the column header "NIWA" for "**VCSR**"

**RC:** (b) Maybe mean magnitude over land to panels?

> **AR:** We considered the suggestion and decided not to add the mean magnitude over land to the panels. The reasons are that the mean magnitude over land is never specifically referenced or discussed in the text and that the panels mainly showcase the high resolution precipitation patterns. Adding text would, furthermore, conceal part of the patterns.

**RC:** Figure 6: Why do the no. samples (circles) differ among the various thresholds in HSS? Explain in the caption.

> **AR:** We added the following sentence to the caption of Figure 6 to explain the reason:
>
> "**At some weather stations no days with $P_{24h}$ > 25 mm and $P_{24h}$ > 50 mm were observed or simulated during certain seasons, therefore no HSS scores could be calculated.**"

**References**

Barstad, I., & Schüller, F. (2011). An extension of Smith's linear theory of orographic precipitation: Introduction of vertical layers. *Journal of the Atmospheric Sciences*, *68*(11), 2695-2709.

Gutmann, E., Barstad, I., Clark, M., Arnold, J., & Rasmussen, R. (2016). The intermediate complexity atmospheric research model (ICAR). *Journal of Hydrometeorology*, *17*(3), 957-973.

Roth, A., Hock, R., Schuler, T. V., Bieniek, P. A., Pelto, M., & Aschwanden, A. (2018). Modeling winter precipitation over the Juneau Icefield, Alaska, using a linear model of orographic precipitation. *Frontiers in Earth Science*, *6*, 20.

Jarosch, A. H., Anslow, F. S., & Clarke, G. K. (2012). High-resolution precipitation and temperature downscaling for glacier models. *Climate Dynamics*, *38*(1-2), 391-409.

Mensh, B., & Kording, K. (2017). Ten simple rules for structuring papers. *PLoS computational biology*, *13*(9), e1005619.

Maraun, D. et al. (2017). Towards process-informed bias correction of climate change simulations. *Nature Climate Change*, *7*(11), 764.

Paeth, H., Pollinger, F., Mächel, H., Figura, C., Wahl, S., Ohlwein, C., & Hense, A. (2017). An efficient model approach for very high resolution orographic precipitation. *Quarterly Journal of the Royal Meteorological Society*, *143*(706), 2221-2234.

Rasmussen, R. et al. (2014). Climate change impacts on the water balance of the Colorado headwaters: high-resolution regional climate model simulations. *Journal of Hydrometeorology*, *15*(3), 1091-1116.

Siler, N., & Durran, D. (2015). Assessing the impact of the tropopause on mountain waves and orographic precipitation using linear theory and numerical simulations. *Journal of the Atmospheric Sciences*, *72*(2), 803-820.

Torma, C., Giorgi, F., & Coppola, E. (2015). Added value of regional climate modeling over areas characterized by complex terrain—Precipitation over the Alps. *Journal of Geophysical Research: Atmospheres*, *120*(9), 3957-3972.

Weidemann, S., Sauter, T., Schneider, L., & Schneider, C. (2013). Impact of two conceptual precipitation downscaling schemes on mass-balance modeling of Gran Campo Nevado ice cap, Patagonia. *J. Glaciol*, *59*(218), 1106-1116.

---

## Author Comment (AC2) · 29 Mar 2019

**Response to Reviewer 2**

**Abbreviations:**

> AR     Author Response (Johannes Horak)
>
> RC     Reviewer Comment

**RC:** This manuscript assesses the added value of ICAR relative to coarse reanalysis for estimating precipitation in complex topography. Not yet widely evaluated, ICAR is a promising tool for a range of applications. The methods in this study are robust and the conclusions are important. The manuscript should be a valuable contribution to the literature.

> **AR:**
>
> We thank the reviewer for his or her time spent on providing valuable and important feedback to the manuscript. We carefully considered all points that were brought up and incorporated the appropriate changes to the manuscript. Please find our detailed responses below.
>
> **Correction to the manuscript independent of the RCs:**
>
> **P5L8:** We found that the list of fields contained in the forcing file was incomplete. We added the two missing fields, the sentence now reads:
>
> "The assembled ICAR forcing file contains ERAI zonal and meridional winds U and V, potential temperature $\Theta$, pressure p, specific humidity $q_v$ , **cloud liquid water mixing ratio $q_c$ , cloud ice water mixing ratio $q_i$** and surface pressure $p_0$ at each 6 h forcing time step and every grid point within the domain."
>
> **P32L14:** The list of employed open-source libraries was incomplete. We added the missing library. The sentence now reads:
>
> "numpy (van der Walt et al., 2011), pandas (McKinney et al., 2010), xarray (Hoyer and Hamman, 2017), matplotlib (Hunter, 2007), cartopy (Met Office, 2010) **and salem (Maussion et al., 2019**)."

**Comments**

**RC:** 1. In terms of the manuscript structure, it seems a bit unusual to have a combined "Methods and Results" section (4). I can see why the manuscript was structured as it is, but I wonder if it could be rationalised at all. Could there be benefits from a more "traditional" separation of methods and results? For example, the major sections could go something like: 1. Introduction 2. Study Area and Data 3. Methods 3.1 ICAR Overview and Setup 3.2 Evaluation Strategy 3.3 Skill Scores and Significance Tests 3.4 Flow Linearity (explaining how flow linearity and stability are calculated) 3.5 Weather Patterns (explaining dataset with figure of weather patterns) 4. Results (as currently structured but removing the methods now described in the previous section) 5. Discussion 6. Conclusions. This is just a possibility; there could be a better way.

> **AR:**

We agree that in this sense the structuring of the manuscript follows a more non-traditional approach. A separation of methods and results has its advantages, however, we are of the opinion that this format lends itself better to manuscripts that focus on one or two central methods. The manuscript in discussion, in comparison, introduces six different methods employed to investigate different aspects of the ICAR simulations. Here, combining methods and results allows for a more fluid reading experience while still enabling non-linear reading where readers may jump from method to method or result to result. This optimizes the logical flow by avoiding zig-zagging as suggested by Mensh, 2017.

**RC:** I would also suggest checking the manuscript for repetition and trying to minimise the amount of "referencing forward" (i.e. sometimes it is not necessary to say "X will be discussed in Section Y").

**AR:** We checked the manuscript for the aforementioned repetitions and removed the forward referencing where possible.

P5L8: We removed the forward reference ""

P7L16-17: We removed the forward reference "."

P12L5-6: We removed the forward reference ""

P13L15-16: We removed the forward reference ""

P27L12: We removed the forward reference ""

**RC:** 2. One specific point on structure is that the optimal model top height is stated in Section 2.3, before the results from the sensitivity test are presented. This should be avoided I think. It is already stated in the relevant part of the results section (i.e. on model top sensitivity) that the remainder of the evaluation uses the optimal model top.

**AR:**

We agree and removed this forward reference. The last line of the paragraph now reads:

"**Therefore, a sensitivity analysis was conducted to identify the optimal elevation of the model top for this study.**"

**RC:** 3. Also regarding the model top sensitivity, it would be interesting to contextualise the variation in ICAR performance shown in Figure 2 by providing the equivalent MSE for ERAI. This could be as a horizontal line on Figure 2 or just stated in the text. I.e., even for the model tops leading to larger errors, do they still outperform ERAI overall?

**AR:**

Figure 2 shows the mean MSE of ICAR and $ICAR_{CP}$ over all alpine weather stations. Indicating the mean MSE of ERAI over all alpine weather stations does not necessarily indicate whether ICAR, on average, outperforms ERAI. To highlight this we added an additional panel to Figure 2 that shows the $SS_{MSE}$ averaged over all alpine weather stations for ICAR and $ICAR_{CP}$.

ICAR is able to outperform ERAI at model top settings of 1.5 km, 2.5 km and 4.0 km, while ICAR$_{CP}$ additional shows added value over ERAI for a model top set at 5.7 km.

The updated Figure 2, the additional panel referenced above is panel b:

[Figure]

**Figure 2.** The average (a) MSE (b) SS$_{MSE}$ of ICAR and ICAR$_{CP}$ time series from simulations for the reference period 2014-2015 at alpine weather stations as a function of the chosen model top (in km above topography). Connecting lines serve as guides to the eye. Panel (c) shows the distribution of skill scores for simulations with a model top set 4.0km above topography at alpine weather stations for the reference period (2014-2015), the full study period (2007-2017) and the reduced study period where the reference period has been removed from the dataset (2007-2013 and 2016-2017). The lower boundary of each box indicates the 25th percentile, the upper boundary the 75th percentile and the dashed horizontal line the mean. Whiskers show the minimum and maximum values of the data set.

**RC:** 4. It is mentioned in the discussion (P29 L19-20) that higher model tops lead to lower precipitation. Does this apply across the full range of model tops tested?

**AR:**

The statement refers to the mean bias observed for simulations with a given model top at alpine weather stations and applies to model tops greater or equal to 1.5 km. The simulations with model tops at 2.5 km and 1.5 km are, on average, wetter than those with a model top at 4 km, while the 0.7 km model top runs are, in that regard, similarly wet with respect to the 4.0 km simulations.

However, due to preliminary work outside of the scope of this article we suspect that the increased wetness associated with lower model tops is due to numerical artifacts. Further research is necessary to converge towards a definitive answer.

**RC:** Would there be any value in adding a second panel to Figure 2 showing mean bias for the different model tops? I.e. given that ICAR is generally low-biased for the Alpine stations, does a 2.5 km model top lead to reduced bias (even if the MSE is little different from 4 km)? Or does ICAR become high-biased with a 2.5 km top?

**AR:**

ICAR is, on average, too dry for all model top settings tested with respect to alpine weather stations. While simulations with 1.7 km and 2.5 km are wettest, preliminary work shows a strong indication that this behavior is due to numerical artifacts associated with the treatment of the model top. Furthermore, as indicated in the discussion (P29L26-27), it is currently not well understood why the precipitation decreases for higher model tops. Therefore, we are of the opinion that an additional panel showing the mean biases for different model tops would (currently) not be of added value.

**RC:** 5. In several places in the manuscript, results are discussed but the corresponding figures/tables are "not shown". This includes the relationship between model top and precipitation magnitude mentioned above, as well as seasonal averages at Alpine station locations (P16 L13-18). The latter case I found confusing initially, as the ERAI seasonality is reasonable-ish in Figure 5 but criticised in the text based on station locations (P16 L17-18). I would suggest considering whether some of the "not shown" figures/tables should be put into a supplement or whether references to them are necessary.

**AR:** While revising the manuscript we greatly reduced the amount of times secondary results are not shown. We added the corresponding Figures to the appendix and added references to them instead of the previously used "not shown". The locations in the manuscript are:

**P10L27, P16L14, P16L17, P27L7, P27L10, P30L2, P30L9**

**RC:** 6. It may be useful to provide the season definitions used in Section 4.5 (i.e. which months).

**AR:** We added abbreviations of the months that are associated with each season to the second paragraph of Section 4.5 (P16L6-10) and the caption of Figure 5. Please note that the updated caption of Figure 5 includes an additional sentence due to another RC (orange, non-bold text):

"The seasonal variations of precipitation as derived from the VCSR data set (Fig. 5b-e) are best reproduced by ICARCP (Fig. 5l-o). However, the improvements over the corresponding ICAR patterns 5g-j) are small and the remainder of this paragraph applies to ICAR and ICARCP alike. When comparing VCSR and ICARCP the similarities are largest for winter (**JJA**, Fig. 5h and 5m) and summer (**DJF**, Fig. 5e and 5o). The differences increase for the remaining seasons, with the Southern Alps being particularly affected. For autumn (**MAM**)**,** VCSR shows the precipitation as below average (Fig. 5b) while ICARCP indicates above average precipitation (Fig. 5l). For spring (**SON**), on the other hand, VCSR shows an increase in precipitation throughout the Southern Alps (Fig. 5d) but ICARCP shows the central part of the Southern Alps as drier than on average (Fig. 5n)."

Figure 5. The top four panels show patterns of P24h averaged over 2007–2016 for VCSR (left), ICAR (second column), ICARCP (third column) and ERAI (right) over the South Island of New Zealand and surrounding ocean. Rows two to five show seasonal deviations of the all-year average patterns, for autumn (**MAM**, second row), winter (**JJA**, third row), spring (**SON**, fourth row) and summer (**DJF**, bottom). Each panel shows the coastline and the 1000 m MSL contour line of the topography. High resolution plots are available in Horak et al. (2018).

**RC:** 7. The panels in Figure 5 are quite small so it is difficult to make out much of the detail. The overall improvement of ICAR over ERAI is clear though. It would be interesting to see a version of the figure zoomed in on the Alpine range, but perhaps this could be in future work.

**AR:** We added higher resolution plots of all the patterns to the data repository (https://doi.org/10.5281/zenodo.1135131) and indicated this in the caption of Figure 5 by adding the sentence:

"**High resolution plots are available in Horak et al. (2018).**"

**RC:** 8. Figure 3b has a spelling error - "coastal".

**AR:** We corrected the spelling error.

**RC:** 9. The boxplots for near-stable conditions in Figure 7c and 7d are quite different. What could be the reason for this?

**AR:** A question of Reviewer 1 led us to the discovery that some gridpoints were erroneously used for the flow linearity analysis. After redoing the analysis Figure 7c and 7d are more similar to each other. However, even in the updated version it is evident that the spread of scores in the "near stable" category is much larger than that in "stable" conditions. This is potentially attributable to the kappa threshold employed to distinguish between near stable and stable atmospheric conditions. If the threshold is set low, days that could reasonably be classified as stable (by investigating potential temperature profiles, for instance) are moved to the near stable category, leading to higher scores there. This behavior is observed for all skill measures employed.

**RC:** 10. There are a few places where the wording and grammar could be a little bit tighter. For example, sometimes "trend" is used when something like "pattern" might be better. There are other examples too, such as the first three sentences of the paragraph beginning on L15 on P20. The manuscript is generally fairly well written, but I would suggest that the authors check the wording and grammar throughout when making revisions.

**AR:** We reread the manuscript and, while making revisions, adjusted the text in cases where wording and grammar seemed problematic. See, for instance:

P3L27: We fixed a spelling error: "To avoid unstable atmospheric condition**s** present in the..."

P12L4: We fixed the grammar and spelling: "at this threshold ICAR$_{CP}$ performs very similar**ly** to ERAI, and that ICAR$_{CP}$ does not improve on modeling the frequency of precipitat**i**on."

P26L12: We exchanged "trend" for "**behavior**"

The paragraph starting at P20L15 now reads: "Of the 4018 days in the eleven-year study period, **1847 fulfill the criteria stated above**. A detailed overview **of** the distribution of these days among the three categories in dependence of κ is given in Table 3. The results from Table 3 ..."

**RC:** 11. In the abstract and discussion it is mentioned that ICAR can reduce MSE by up to 53%. If this is the maximum reduction, what is the mean/median? This may be worth including to give the "overall" picture.

**AR:** We rephrased the sentence in the abstract and discussion. In the abstract the rephrased sentence now reads:

"Furthermore, ICAR is found to provide added value over its ERA-Interim reanalysis forcing data set for alpine weather stations, improving mean squared errors (MSE) **by up to 53 % and 30 % on median.**"

In the discussion the updated sentence now is:

"At alpine sites in complex topography ICAR$_{CP}$ is then able to reduce mean squared errors in comparison to its ERAI forcing dataset by up to 53 % **and 30 % on median.**"

**RC:** 12. It could be mentioned again in the discussion/conclusion that a comparison of ICAR and WRF (or a similar model) might also be interesting for this study area. This might help us to understand some of the possible factors limiting ICAR performance discussed in Section 5. It would also give an idea of the relative performance gain from using WRF (if any) in a different climatic context to that tested in Gutmann et al. (2016).

**AR:** We agree that this might be of interest. However, ICAR is still a relatively new model. Preliminary work outside of the scope of this article gives us reason to suspect that other factors, such as numerical artifacts at the model top, might currently limit or influence the performance of ICAR. Before a meaningful comparison to a dynamic downscaling model can be made, it is necessary to develop a better understanding of these issues and how they can be overcome (or avoided) in future versions of ICAR.

**References**

Mensh, B., & Kording, K. (2017). Ten simple rules for structuring papers. *PLoS computational biology*, *13*(9), e1005619.

---

## Author Comment (AC3) · 29 Mar 2019

**Response to Reviewer 3**

**Abbreviations:**

AR      Author Response (Johannes Horak)

RC      Reviewer Comment

**RC:** Overview This well-written manuscript details a comparison between ERA-Interim and ICAR at generating precipitation over New Zealand's south island. They find that ICAR adds value over ERA Interim at most alpine locations, but not at coastal stations. They additionally tease apart ICAR performance during different flow regimes (identified by the Froude number) and during different weather regimes (identified through synoptic patterns). The work is useful and complete, and I have only minor comments, enumerated below.

> **AR:**
>
> We thank the reviewer for her or his time and the detailed comments and criticism of our manuscript! We reflected on each point and modified the manuscript accordingly, please find the detailed answers below!
>
> **Correction to the manuscript independent of the RCs:**
>
> **P5L8:** We found that the list of fields contained in the forcing file was incomplete. We added the two missing fields, the sentence now reads:
>
> "The assembled ICAR forcing file contains ERAI zonal and meridional winds U and V, potential temperature $\Theta$, pressure p, specific humidity $q_v$ , **cloud liquid water mixing ratio $q_c$ , cloud ice water mixing ratio $q_i$** and surface pressure $p_0$ at each 6 h forcing time step and every grid point within the domain."
>
> **P32L14:** The list of employed open-source libraries was incomplete. We added the missing library. The sentence now reads:
>
> "numpy (van der Walt et al., 2011), pandas (McKinney et al., 2010), xarray (Hoyer and Hamman, 2017), matplotlib (Hunter, 2007), cartopy (Met Office, 2010) **and salem (Maussion et al., 2019**)."

**Specific Comments**

**RC:** P. 3, l. 27-29: During my first read through of the manuscript this sentence made me question how this replacement of unstable locations/times with weakly stable locations/times impacts ICAR's performance (since it's very unphysical). Some comment here or perhaps in the introduction about application of ICAR during unstable conditions (referring to section 2.6, which is how it is handled in this manuscript), and how/where this factor limits ICAR's use, is warranted.

> **AR:** We agree that an analysis of how ICAR performs under unstable/near stable conditions is necessary. For this reason, we conducted a detailed analysis of ICAR performance in dependence of the flow regime and atmospheric stability in Section 4.6. To avoid unnecessary zig-zagging as suggested by Mensh (2017) we did not include a forward reference since the Abstract and

Introduction both mention the conducted analysis and the corresponding results (P1L14 and P3L6).

We also corrected the erroneously given value of $10^{-7}$ s$^{-1}$ at P3L29 for the lower limit of N. The correct value now shown in the manuscript is **3.2 $10^{-4}$ s$^{-1}$**.

**RC:** P. 5, l. 10: '6 h h' the second h is a mistake

    **AR:** We removed the additional h

**RC:** P. 5, l. 21-24: I found the way this is notated to be somewhat confusing. I think the reason the authors are using the nomenclature 'ICARcp' to replace P(t) (i.e., ICAR precipitation added to ERA Interim convective precipitation regridded through bilinear interpolation to the 4km grid) is because it's basically ICAR plus convective precipitation. But this seems more complicated than necessary – why not use P(t) and Pi(t) throughout the text? If the authors insist on keeping ICARcp and ICAR then they should use this nomenclature in equation 1 and include a sentence explaining the nomenclature after the equation.

    **AR:** We employed the variables $P(t)$, $P_{CP}(t)$ and $P_I(t)$ in equation (1) to conform to Journal guidelines where the use of multi-letter variables is discouraged where possible. (see Section Mathematical requirements, Symbols and Equations, index b). However, the nomenclature ICAR$_{CP}$ and ICAR was chosen to allow a reader skipping parts of the introduction to immediately identify the data source of a time series or precipitation map. Additionally, due to the length of the manuscript, choosing the nomenclature ICAR$_{CP}$ over $P(t)$ and ICAR over $P_I(t)$ avoids having to reestablish the variable definition all over again to remind the reader of the meaning of the variable.

**RC:** P. 7 l. 12: 'In case of the coastal weather stations,...' is awkward.

    **AR:** We rephrased the sentence. Please note that the corrected version includes another adaption due to a comment by reviewer 4 (orange, non-bold text). The sentence now reads: "**At** coastal weather stations, records from the New Zealand National Climate Database (NCD, https://cliflo.niwa.co.nz) were employed."

**RC:** P. 8, caption of Table 1, last sentence: 'north respectively south' should read 'north and south, respectively'

    **AR:** We rephrased the sentence to: "Δ was not considered for coastal weathers stations and no values were assigned for Mahanga and Larkins since they lie **north and south, respectively,** of the main alpine crest."

**RC:** P. 12, l. 4: 'performs very similar' should read 'performs very similarly'

    **AR:** We rephrased accordingly and fixed a spelling error in precipitation: "Since only small negative scores are found and the median score is 0.01 for all alpine stations, this indicates, that at this threshold ICAR$_{CP}$ performs very similar**ly** to ERAI, and that ICAR$_{CP}$ does not improve on modeling the frequency of precipitat**i**on."

**RC:** P. 13: Fig 3 panel b: coastal is misspelled in title.

    **AR:** We corrected the spelling.

**RC:** P. 14, table 2 caption, last sentence, asterisk is misspelled.

    **AR:** We corrected the spelling.

**RC:** P. 16, l. 16-17: It's unclear to me exactly what this sentence is describing since the figure is not shown; does this mean that the amplitude of the seasonal cycle is too small in ICAR or more generally that ICAR underestimates climatological precipitation at some locations? More discussion is warranted and perhaps this figure should be included in the manuscript.

    **AR:**

We included an additional figure and rephrased for clarity. P16L4 now more specifically indicates that this paragraph is about the seasonal precipitation patterns, while the paragraph starting at P16L13 is concerned with results at the weather station level. Answering the reviewer's questions: Both statements are true for the alpine weather stations investigated. On average ICAR$_{CP}$ understimates the amplitude of the seasonal cycle and the climatological precipitation at the weather stations situated in the Southern Alps. However, in contrast to ERAI, which predicts spring to be the wettest season and autumn as the driest, ICAR$_{CP}$ is able to reproduce the characteristics of the measured seasonal cycle (e.g., winter as the driest season and summer and spring similarly wet).

**P16L4 now reads:** "The seasonal variations of precipitation **patterns** as derived from the VCSR data set (Fig. 5b-e) are best reproduced by ICAR$_{CP}$ (Fig. 5l-o)."

**P16L13 now includes a reference to the additional Figure added to the Appendix instead of 'not shown':** "Seasonal averages of daily accumulated precipitation $\overline{P_{24h}}(\text{se})$ derived from measurements at the alpine weather stations show winter as the driest season, summer as the wettest and the transitional seasons in between (**see Fig. A1**)."

**Additional Figure and caption:**

[Figure]

**Figure A1.** Box and whisker plot of the mean daily precipitation (y-axis) for each season $\overline{P_{24h}}(\text{se})$ at alpine weather stations as measured or calculated by ICAR$_{CP}$ and ERAI (x-axis). All amounts were calculated using the entire $P_{24h}$ time series available at each weather station. The lower boundary of the box indicates the 25th percentile, the upper boundary the 75th percentile and the horizontal dashed line the mean. Whiskers show the minimum and maximum values of the data set.

**RC:** P. 16, l 23-24: Is there any reason to think that the correspondence in seasonal errors between ICAR and ERA-I (i.e., that both have largest errors in summer and smallest in winter) is causal? That is, since ERA-Interim provides lateral boundary conditions for ICAR?

> **AR:** While this is a possibility we are of the opinion that, at least for winter, this correspondence is mostly due to a characteristic of the mean squared error. E.g. since ICAR and ICAR$_{CP}$ generally underestimate measured precipitation it follows that the potential magnitude of the MSE is reduced for winter since it is the driest season. During summer, however, it seems feasible that convective events that are missed by ICAR (which cannot model convective precipitation) and by ERAI alike (which potentially misses them due to the coarse grid spacing) contribute to the increase. This would apply to ICAR$_{CP}$ as well since it incorporates convective precipitation from ERAI.

**RC:** P. 17, Figure 5: masking the ICAR and ERAI values over the ocean would be less distracting (since there is no 'truth' over the ocean, anyways).

> **AR:** While it is not possible to compare ICAR precipitation over the ocean to the VCSR, we still see value in showing the precipitation patterns over the ocean. It showcases the behavior of the model there, i.e. that precipitation is indeed generated even though there is no topography present. To reference this behavior and the choice not to mask values above the ocean in the manuscript we added an additional sentence to the paragraph starting at P15L8:
>
> "**While above the ocean no data is available for the VCSR, the results clearly show that ICAR is able to generate precipitation with seasonal variation above the ocean where no topography is present (Fig. 5f-j).**"

**RC:** P. 19, l. 11: What percentage of the crest of the southern Alps is over 1500m?

> **AR:** If the elevations of all points used for the definition of the Alpine Crest in Figure 1 are extracted from the SRTM digital elevation model (3 arcsecond grid-spacing), approximately 97% of the crest lie above an elevation of 1500 m MSL.

**RC:** Based on Fig. 1 it seems closer to 1000m would be a somewhat more appropriate height to use in the calculation of Froude number;

> **AR:** The average elevation of the Southern Alps is 1100 m MSL if the area east and west of the alpine crest up to a distance of $0.5°$ is averaged over (corresponding to the approximate width of the Southern Alps of 60 km referenced in Section 3.1).

**RC:** are the results pertaining to the Fr<1 and Fr>1 cases sensitive to this mountain height?

> **AR:** Choosing a lower value for H would shift days from the Fr < 1 regime to the Fr >= 1 regime (see equation 4) and vice-versa for higher values of H. The observed characteristics of ICAR remain the same even if instead of H = 1500 m, H = 1000 m or H = 1750 m is chosen. However,

for H = 1750 m the number of cases in the Fr >= 1 regime is too low to calculate meaningful scores.

**RC:** P. 25, second sentence in Fig 9 caption: This sentence is poorly worded.

**AR:** We reworded the first two sentences in the caption: "Box and whisker plot of $SS_{MSE}$ calculated for all alpine weather station in dependence of the synoptic weather pattern (**x-axis;** Kidson, 2000). **The regime associated with each weather pattern is indicated by color shadings in the lower part of the plot.**"

**RC:** P. 26, l. 12-13: This sentence is poorly worded.

**AR:** We rephrased and split the sentence in two: "**Furthermore, at Ivory, the trend found in the measurements is correctly reproduced by $ICAR_{CP}$ and ERAI. The absolute amounts of precipitation are, while underestimated, better modeled by $ICAR_{CP}$.**"

**RC:** P. 29. L18-23: Can the authors speculate why there is this sensitivity to model top height?

**AR:** The authors are currently investigating this sensitivity and hope to present answers in a follow-up study. We currently speculate that the behavior may be caused by divergences and convergences in the forcing wind field (see Section 5. discussion P31L10-12) and, at lower model top settings, by numerical artifacts due to the way the model top is treated. However, further research is necessary.

**RC:** P. 29, L. 21: 'estimation the model' is missing 'of'

**AR:** We inserted the missing 'of'.

**RC:** P. 30, L. 9-13: This paragraph should be expanded for clarity (i.e., rather than saying 'solution to issue (iv) it would be helpful to repeat the description of the issues).

**AR:** We restated issue (iv) in the referenced paragraph (P31L9-13) and added reference to the relevant Figures.

"**At a model top setting of 4 km above topography, seeder-feeder interaction between synoptic clouds and orographically lifted moist air may mostly be eliminated. Increasing the model top is an apparent solution to this issue.** However, the sensitivity study in Sect. 4.3 showed, that this does not lead to a decrease in the MSE of ICAR or $ICAR_{CP}$ **(Fig. 2a), nor does it increase model skill for time series at the alpine weather stations (Fig. 2b).**"

**References**

Mensh, B., & Kording, K. (2017). Ten simple rules for structuring papers. *PLoS computational biology*, *13*(9), e1005619.

---

## Author Comment (AC4) · 29 Mar 2019

**Response to Reviewer 4**

**Abbreviations:**

AR     Author Response (Johannes Horak)

RC     Reviewer Comment

**RC:** I have not been able to complete a proper review of this manuscript due to other more pressing time commitments. In lieu of a full review, I have a few minor comments that can be easily addressed by the authors:

**AR:**

We thank the reviewer for the comments on the manuscript. We went through them and adapted the manuscript accordingly.

**Correction to the manuscript independent of the RCs:**

**P5L8:** We found that the list of fields contained in the forcing file was incomplete. We added the two missing fields, the sentence now reads:

"The assembled ICAR forcing file contains ERAI zonal and meridional winds U and V, potential temperature $\Theta$, pressure p, specific humidity $q_v$ , **cloud liquid water mixing ratio $q_c$ , cloud ice water mixing ratio $q_i$** and surface pressure $p_0$ at each 6 h forcing time step and every grid point within the domain."

**P32L14:** The list of employed open-source libraries was incomplete. We added the missing library. The sentence now reads:

"numpy (van der Walt et al., 2011), pandas (McKinney et al., 2010), xarray (Hoyer and Hamman, 2017), matplotlib (Hunter, 2007), cartopy (Met Office, 2010) **and salem (Maussion et al., 2019**)."

**Specific Comments**

**RC:** 1. The VCS gridded data set is a thin-plate spline based gridded observation set using a mean rain surface as a covariate. This surface is derived using "expert judgement" from observations and elevation, not from "physics-based regional climate modelling" as stated on line 3 page 7.

**AR:** We rephrased the corresponding sentence accordingly, it now reads:

"While the VCSR does not necessarily represent the actual distribution of precipitation (Tait et al., 2012), and may miss precipitation events (Tait and Turner, 2005), it serves as an approximation to an observational gridded dataset and is based on observations and **expert judgement**."

**RC:** 2. For the NCD database, a link https://cliflo.niwa.co.nz could be included.

**AR:** We included a link to the NCD database. The sentence at P7L12 now reads: "At coastal weather stations, records from the New Zealand National Climate Database (NCD, https://cliflo.niwa.co.nz) were employed. "

**RC:** 3. The caption for Figure 4 has the elevation of Albert Burn as 120m it should be 1280m.

**AR:** We corrected the caption of Figure 4.

---

## Referee Report (RR1)

The authors have carefully the revised the manuscript and addressed all of the points in my first review. I found only a very small number of minor points to consider, but I think the manuscript is suitable for publication.

P1 L13: Consider slightly rewording clause from "improving mean squared errors (MSE) by up to 53% and 30% on median" to "improving mean squared errors (MSE) by 30% on average (median) and up to 53%". I think "on median" sounds a bit unnatural.

P2 L21: Insert comma between "measurements" and "with"

P7 L18: Change "stations: A" to "stations: a"

P9: I accept the authors' response about retaining the manuscript structure. The only further point to consider (possibly) is whether sections 4.1 and 4.2 could be split out into an "evaluation strategy" (or similar) section, with the remainder of "methods and results" (section 4.3 onwards) becoming its own section? This could perhaps help to separate things a little bit, but I think it is up to the authors (i.e. it is not a big issue).

P11 L5: Consider adjusting wording from "equivalent to a 30% reduction of error on median relative to…" to "equivalent to a median reduction of error of 30% relative to…"

P31 L25-28: Could this point be included instead in the first paragraph on P30 (specifically point "iv", beginning on L9)? Otherwise the final paragraph of the discussion on P31 seems a little bit "tacked on".

---

## Author Response (AR2)

**Response to the Editor**

**Abbreviations:**

AR     Author Response (Johannes Horak)

EC     Editor Comment

**EC:** If possible, I suggest avoiding using acronyms (e.g. ERAI, T, TNW, W, etc.). Some of the readers will likely read the conclusions before reading the entire manuscript.

**AR:** We adapted the conclusions accordingly. The new version now reads:

"In this study, simulations with ICAR were found to provide added value over ERA-**Interim** for 24 h accumulated precipitation on the South Island of New Zealand for alpine weather stations. In contrast to the almost consistently positive results found for the alpine weather stations, ICAR provides no added value over ERA-**Interim** for coastal weather stations. A comparison of average and seasonal precipitation patterns of an operational gridded rainfall data set with ICAR showed good agreement. Grouping the available data according to Froude number revealed that stable atmospheric conditions with higher degree of flow linearity lead to higher skill scores, and that ICAR provides added value over ERA-**Interim** even for days with near-stable conditions and stable days with lower flow linearity. A grouping according to the synoptic situation showed that values of $SS_{MSE}$ are generally high for weather patterns associated with flow approximately perpendicular to the alpine range and lowest for weather patterns exhibiting flow parallel to the Southern Alps . While ICAR in principle does not require observations to be calibrated, the model top for this study was determined with a sensitivity analysis. All other settings could be adopted from default. With the adjusted model top, however, consistent added value for stations in complex topography was found, with a reduction of the median error by 30%. Clear improvement may be expected on further site-specific calibration to observations as routinely performed in regional climate model based downscaling. Further research on how ICAR fields are influenced by the forcing data set will be necessary. "

**EC:** [Figures 8 and 11] Consider moving to the Supplementary Material.

**AR:** We moved Figures 8 and 11 to the Supplementary Material.

**EC:** [Appendix A and Figures A1 to A4] Please move the appendix to the Supplementary Material, i.e. have them as a separate file and not part of the main text.

**AR:** All Figures from the Appendix have been move to the Supplementary Material.

**Response to Referee 2**

**Abbreviations:**

AR     Author Response (Johannes Horak)

RC      Referee Comment

**RC:** The authors have carefully the revised the manuscript and addressed all of the points in my first review. I found only a very small number of minor points to consider, but I think the manuscript is suitable for publication.

    **AR:** The Authors once again want to express their gratitude to the referee for taking the time to review the manuscript in great detail and for making valuable comments and suggestions.

**RC:** P1 L13: Consider slightly rewording clause from "improving mean squared errors (MSE) by up to 53% and 30% on median" to "improving mean squared errors (MSE) by 30% on average (median) and up to 53%". I think "on median" sounds a bit unnatural.

    **AR:** We reworded the corresponding sentence. It now reads:
    "Furthermore, ICAR is found to provide added value over its ERA-Interim reanalysis forcing data set for alpine weather stations, improving mean squared errors (MSE) by 30% on **average (median) and up to 53%**"

**RC:** P2 L21: Insert comma between "measurements" and "with"

    **AR:** We inserted the comma.

**RC:** P7 L18: Change "stations: A" to "stations: a"

    **AR:** We corrected to the suggested version.

**RC:** P9: I accept the authors' response about retaining the manuscript structure. The only further point to consider (possibly) is whether sections 4.1 and 4.2 could be split out into an "evaluation strategy" (or similar) section, with the remainder of "methods and results" (section 4.3 onwards) becoming its own section? This could perhaps help to separate things a little bit, but I think it is up to the authors (i.e. it is not a big issue).

    **AR:** While the authors agree that this is a possible way to restructure the manuscript, in our opinion the evaluation strategy as well as the description of the scores and significance tests (subsections 4.1 and 4.2) are still part of the papers methodology and therefore fit into the section "methods and results".

**RC:** P11 L5: Consider adjusting wording from "equivalent to a 30% reduction of error on median relative to…" to "equivalent to a median reduction of error of 30% relative to…"

    **AR:** We adjusted the wording accordingly, it now reads:

    "For the alpine weather stations, values of $SS_{MSE}$ calculated across the entire period when data is available (see Table 1 for details) indicate a median $SS_{MSE}$ of 0.3, **equivalent to a median reduction of error of 30% relative to** ERAI for locations in complex alpine topography."

**RC:** P31 L25-28: Could this point be included instead in the first paragraph on P30 (specifically point "iv", beginning on L9)? Otherwise the final paragraph of the discussion on P31 seems a little bit "tacked on".

**AR:** We agree with this assessment. We removed P31 L25-28 and adapted P30 L9-12 accordingly:

**P30 L9-12**:

[revised manuscript text omitted]